# Compromised DNA repair is responsible for diabetes-associated fibrosis

Varun Kumar[1,2,3,*] , Raman Agrawal[4], Aparamita Pandey[1], Stefan Kopf[1,3], Manuel Hoeffgen[1], Serap Kaymak[1], Obul Reddy Bandapalli[5,6], Vera Gorbunova[7], Andrei Seluanov[7], Marcus A Mall[4,8,9], Stephan Herzig[3,10,11,12] & Peter P Nawroth[1,3,10,11,**]

## Abstract

Diabetes-associated organ fibrosis, marked by elevated cellular senescence, is a growing health concern. Intriguingly, the mechanism underlying this association remained unknown. Moreover, insulin alone can neither reverse organ fibrosis nor the associated secretory phenotype, favoring the exciting notion that thus far unknown mechanisms must be operative. Here, we show that experimental type 1 and type 2 diabetes impairs DNA repair, leading to senescence, inflammatory phenotypes, and ultimately fibrosis. Carbohydrates were found to trigger this cascade by decreasing the $NAD^+$/NADH ratio and NHEJ-repair *in vitro* and in diabetes mouse models. Restoring DNA repair by nuclear overexpression of phosphomimetic RAGE reduces DNA damage, inflammation, and fibrosis, thereby restoring organ function. Our study provides a novel conceptual framework for understanding diabetic fibrosis on the basis of persistent DNA damage signaling and points to unprecedented approaches to restore DNA repair capacity for resolution of fibrosis in patients with diabetes.

**Keywords** diabetes; DNA double-strand breaks; nuclear isoform of the Receptor for Advanced Glycation End products; pulmonary fibrosis; reducing carbohydrates
**Subject Categories** DNA Replication, Recombination & Repair; Metabolism; Molecular Biology of Disease
**The EMBO Journal (2020) 39: e103477**

## Introduction

Maladaptive metabolic reprogramming is associated with severe organ dysfunction (Maynard *et al*, 2015). Metabolic changes that occur in complex metabolic disorders such as diabetes are associated with the inability to regenerate the damaged organs (lung, kidney, liver, eye, or nerve) (Maynard *et al*, 2015; Kopf *et al*, 2018; Touat *et al*, 2018). Conversely, neonatal organs are capable of coping with the similar situation in a more accurate way than adults, by timely "turning-on" their repair systems (Charron & Bonner-Weir, 1999; Dobbin *et al*, 2013; Bhatt *et al*, 2015; Wu *et al*, 2016) (Rudolph & Heyman, 1974). Furthermore, low energy-demanding intrauterine life relies upon ROS free anaerobic glycolysis (Rudolph & Heyman, 1974; Fisher *et al*, 1980; Lopaschuk *et al*, 1992; Mitchell & Van Kainen, 1992; Reynolds *et al*, 1996; Webster & Abela, 2007). However, soon after birth, the developmental process requires much more energy than intrauterine life. The postnatal oxygen-rich environment leads to abrupt mitochondrial replication and increased energy production via oxidative phosphorylation (Turrens, 1997, 2003). This sudden and remarkable change is sufficient to meet the energy demands of the growing organs (Turrens, 1997, 2003; Puente *et al*, 2014), but at the cost of accumulation of reactive by-products, such as ROS. These reactive by-products are known to damage various cellular components such as proteins, lipids, and DNA (Moos *et al*, 2000; Marnett *et al*, 2003; Hoeijmakers, 2009). These sudden metabolic changes must, therefore, be balanced by the timely adaptation of the newborn to develop an effective tissue repair system.

In adults, modern lifestyle is accompanied by severe metabolic changes that predispose them to accelerated aging, marked by an

1 Department of Medicine I and Clinical Chemistry, University Hospital of Heidelberg, Heidelberg, Germany
2 European Molecular Biology Laboratory, Advanced Light Microscopy Facility, Heidelberg, Germany
3 German Center for Diabetes Research (DZD), Heidelberg, Germany
4 Department of Translational Pulmonology, Translational Lung Research Center Heidelberg (TLRC), German Center for Lung Research (DZL), University of Heidelberg, Heidelberg, Germany
5 Hopp Children's Cancer Center, Heidelberg, Germany
6 Medical Faculty, Heidelberg University, Heidelberg, Germany
7 Department of Biology, University of Rochester, Rochester, NY, USA
8 Department of Pediatric Pulmonology, Immunology and Critical Care Medicine, Charité - Universitätsmedizin Berlin, Berlin, Germany
9 Berlin Institute of Health (BIH), Berlin, Germany
10 Institute for Diabetes and Cancer, Helmholtz Center Munich, Neuherberg, Germany
11 Joint Heidelberg-IDC Translational Diabetes Program, Helmholtz-Zentrum, München, Germany
12 Technical University Munich, Munich, Germany
*Corresponding author: Tel: +49 6221 56 6960; E.mail: varun.kumar@med.uni-heidelberg.de
**Corresponding author: Tel: +49 6221 56 8601; E-mail: peter.nawroth@med.uni-heidelberg.de

increased risk of various chronic diseases (Gluckman *et al*, 2007, 2009; Aagaard-Tillery *et al*, 2008; Harron *et al*, 2011; Maynard *et al*, 2015). Thus, the health risks, associated with a Western lifestyle, are the net result of defective defense and repair mechanisms, affecting kidney, lung, liver, heart, brain, nerve, eye, and vessels (Gluckman *et al*, 2009; Maynard *et al*, 2015). These retrogressive metabolic changes might especially be valid for both type 1 and type 2 diabetes, known to be associated with increased levels of geno-toxic ROS and dicarbonyls (Aagaard-Tillery *et al*, 2008). However, it remains unknown, whether increased damage by reactive metabo-lites alone, or in combination with impaired DNA repair, is responsi-ble for diabetes-associated organ fibrosis.

A defective DNA repair potential leads to cellular senescence, pro-inflammatory Senescence-Associated Secretory Phenotype (SASPs), and ultimately fibrosis (Toussaint *et al*, 2002; Narita *et al*, 2006; Rodier *et al*, 2009; Freund *et al*, 2010; O'Driscoll, 2012; Cheresh *et al*, 2013; Chilosi *et al*, 2013; Tchkonia *et al*, 2013; Pove-dano *et al*, 2015; Kumar *et al*, 2017; Schafer *et al*, 2017, 2018; Keijzers *et al*, 2018; Rivera *et al*, 2018). Moreover, increased DNA damage, senescence, and persistent DNA damage signaling have been described in type 1 and type 2 diabetes (Lorenzi *et al*, 1986; Blasiak *et al*, 2004; Tatsch *et al*, 2012; Bhatt *et al*, 2015; Palmer *et al*, 2015; Burton & Faragher, 2018; Kopf & Nawroth, 2018; Kopf *et al*, 2018). However, organs such as kidney, lungs, and liver show the highest incidence of fibrosis and other complications (Ban & Twigg, 2008; Kopf & Nawroth, 2018; Kopf *et al*, 2018; Tala-katta *et al*, 2018). Yet, it remains unclear whether and how an impaired DNA repair system is responsible for the development of organ dysfunction in diabetes. Recent data indicate that a defective or non-functional DSBs repair system is linked to diabetic compli-cations. This signifies the importance of DNA repair in diabetes (Kornum *et al*, 2007; Ban & Twigg, 2008; Honiden & Gong, 2009; Yang *et al*, 2011; Giovannini *et al*, 2014; Tornovsky-Babeay *et al*, 2014; Bhatt *et al*, 2015; Russo & Frangogiannis, 2016). Besides, several clinical studies indicate that radiation pneumonitis is more common in patients with diabetes and lung cancer (Kornum *et al*, 2007; Barnett *et al*, 2009; Busaidy *et al*, 2012; Tornovsky-Babeay *et al*, 2014). Similarly, skin necrosis and fibrosis are also more frequent in diabetic patients undergoing radiation therapy against cancer (Peairs *et al*, 2011; Busaidy *et al*, 2012; Jha *et al*, 2016; Kalman *et al*, 2018). Enhanced sensitivity of the microvasculature to radiation has also been demonstrated in patients with diabetes (Clark *et al*, 2002; Simo *et al*, 2018). The increased sensitivity toward radiation injury, fibrosis, and the poor clinical outcome is suggestive of a yet unknown mechanism accounting for inefficient DNA repair in diabetes (Kornum *et al*, 2007; Ban & Twigg, 2008; Honiden & Gong, 2009; Yang *et al*, 2011; Giovannini *et al*, 2014; Tornovsky-Babeay *et al*, 2014; Bhatt *et al*, 2015; Russo & Frangogiannis, 2016).

Recently, the central role of nuclear RAGE in DNA repair, senes-cence, and remission of fibrosis has been described (Kumar *et al*, 2017). RAGE is constitutively expressed during the first postnatal term (Brett *et al*, 1993), whereas in adults its abundant expression is confined to lungs (Brett *et al*, 1993; Shubbar *et al*, 2012), neuronal, and other cells (Vlassara *et al*, 1994; Abel *et al*, 1995; Muller-Krebs *et al*, 2008). Moreover, the radiation-mediated DSBs can induce the expression of RAGE in all organs tested (Bucciarelli *et al*, 2006; Kumar *et al*, 2017).

The hypothesis of this study, therefore, was that a maladaptive metabolic stress in diabetes results in impaired DNA repair. By using nuclear phosphomimetic RAGE (RAGE$^{S376E–S389E}$), but not the non-phosphorylated RAGE (RAGE$^{S376A–S389A}$), a tool to overcome diabetes-associated DNA damage, senescence, persistent DNA damage signaling, and inflammation, we can provide firm evidence that hyperglycemia overrides the NHEJ-repair potential and that the diabetes-dependent organ fibrosis in kidney and lung can be reduced by improving DNA repair. Thus, diabetes is a form of maladaptive metabolic reprogramming, associated with impaired DNA-DSBs repair. Overriding the diabetes-induced defective DNA repair is a novel therapeutic approach for the reduction of diabetes-associated fibrosis in lung and kidney.

# Results

## Postnatal metabolic reprogramming of lungs is associated with elevated ROS exposure as well as increased DNA repair potential

To understand the role of high $O_2$ tension and mitochondrial ener-getics in DNA damage, mitochondrial DNA was quantified in lung tissue obtained directly after birth and the subsequent postnatal days. Exposure of the lung to the environmental $O_2$ concentration resulted in an increase in mitochondrial DNA (Fig 1A and B), ROS formation (Fig 1C and Appendix Fig S1A), and a slight elevation in inflammation marker IL-6 (Appendix Fig S1B). Increased ROS, known to affect the integrity of the genome (Turrens, 1997; Cheresh *et al*, 2013), leads to unwanted DNA modifications (Fig 1D and E; left panel and 1F) as well as other DNA damages. Congruently, DSBs signaling was markedly enhanced with increased mitochon-drial DNA content (Figs 1B, D and E; right panel, 1G and EV1A and B). Furthermore, this initial ROS-mediated DNA damage was also associated with an initial decrease in the cell proliferation, as deter-mined by the cell division marker pH3-S10 (Fig EV1C and D). These changes were most prominent on day 3, and after that, cellular reprogramming was able to induce sufficient repair capacity to ensure a healthy life.

Moreover, the increased repair capacity correlated well with the continually increasing $NAD^+/NADH$ ratio from postnatal day 1 to day 14 (Fig 1H). Thus, the most dramatic change in metabolism, the sudden exposure to environmental oxygen, is accompanied by increased mitochondrial content; at the beginning, increased DNA damage and subsequently a sustained change in cellular defense pathways involving pATM triggered DNA repair. Thus, the question arises whether disorders of similar pathways explain DNA damage, SASP, and fibrosis in diabetes.

## Exposure to increasing concentrations of reducing carbohydrates impairs cellular DNA repair

An increased concentration of reducing sugars is known to be associ-ated with increased oxidative stress and ROS formation (Lorenzi *et al*, 1986; Muller-Krebs *et al*, 2008; Tatsch *et al*, 2012). Thus, to simulate the clinical state of hyperglycemia, human alveolar type II cells (A549 cells) were cultured under either normal glucose (Fig 2A; left panel), high glucose (Fig 2A; right panel), ribose (Fig EV2A), or fructose (Fig EV2B) supplemented media as described previously

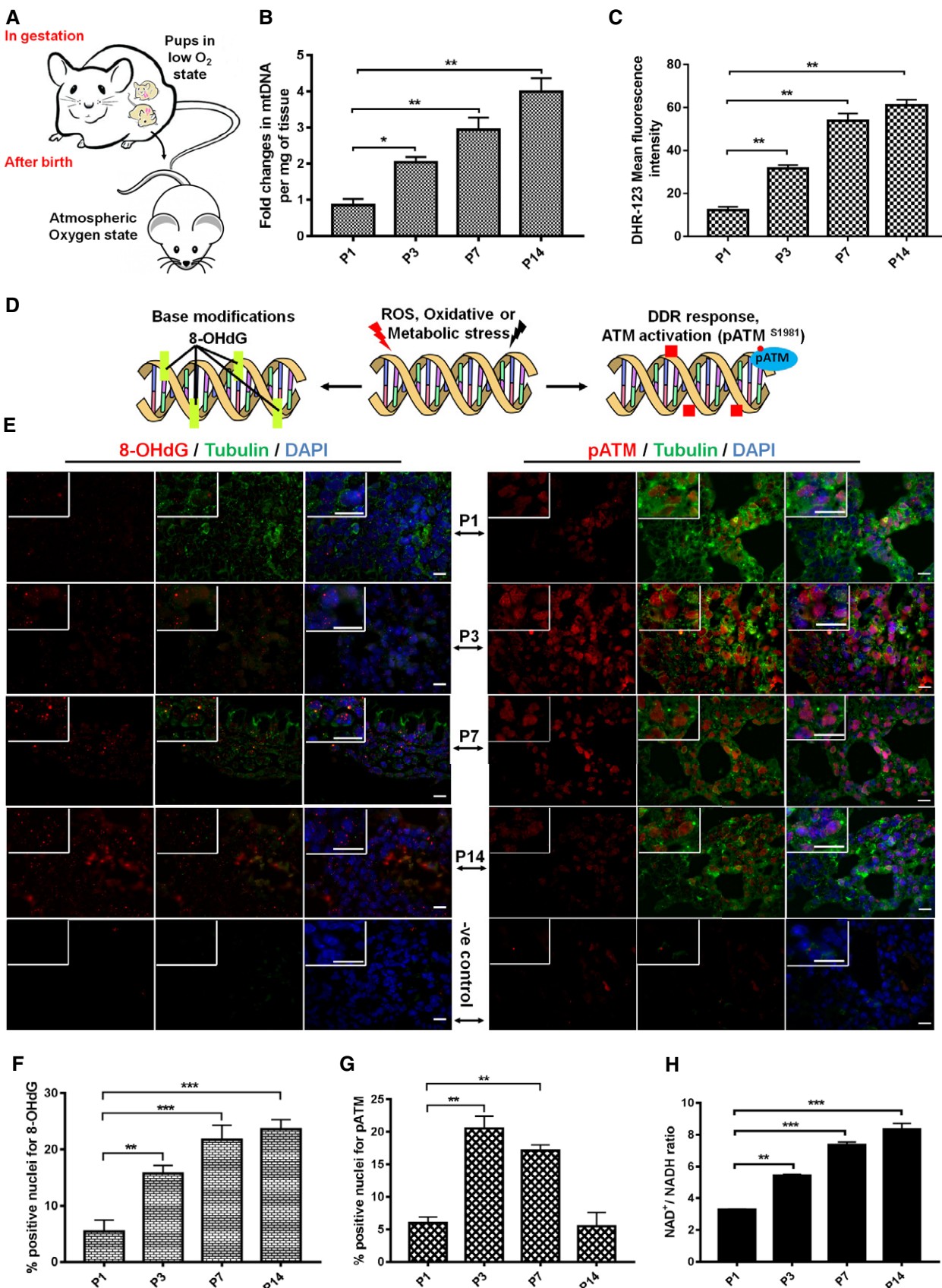

Figure 1.

◄

**Figure 1.  Increased pO$_2$, mitochondrial respiration, and oxidative stress lead to the activation of immediate DNA damage response after parturition of normal live pups.**

A  Schematic representation of the oxygenic state of pre- and postnatal intrauterine life of mice pups.

B  Quantitative PCR analysis of postnatal lungs for mitochondrial DNA (mtDNA). Actin was used as a normalization control. (mean $\pm$ SD, *$P$ < 0.05; **$P$ < 0.01; $N$ = 3).

C  Quantitative analysis of oxidative stress in postnatal pups of indicated age (in days) using the peroxide linked oxidative stress marker DHR-123, as described in Methods (mean $\pm$ SD, **$P$ < 0.01; $N$ = 8).

D  Schematic representation of DNA damage (oxidative or DSBs) on naive DNA.

E  Immunofluorescence analysis of oxidative DNA damage (left panel) and the DNA double-strand breaks sensor kinase marker pATM (right panel) in postnatal murine lungs. Here, anti-8-oxo-7, 8-dihydroguanine was used for the detection of oxidative DNA modification (8-oxoG, left panel), and anti-pATM Serine 1981 was used for the analysis of the activation of the DDR (pATM, right panel). Green cytoplasmic staining represents $\alpha$-tubulin; blue DAPI staining represents nucleus (scale 10 $\mu$m). No primary antibody control served as a negative control (P14) of the staining.

F  Mean percentage of nuclei positive for oxidative DNA damage marker 8-oxoG in postnatal murine lungs, as determined by immunofluorescence analysis (mean $\pm$ SEM, **$P$ < 0.01; ***$P$ < 0.001; $N$ = 8).

G  Mean percentage of nuclei positive for DNA damage marker pATM in postnatal murine lungs, as determined by immunofluorescence analysis (mean $\pm$ SEM, **$P$ < 0.01; $N$ = 8).

H  Quantitative analysis of the NAD$^+$/NADH ratio in postnatal lungs after homogenization in extraction buffer (mean $\pm$ SD, **$P$ < 0.01; ***$P$ < 0.001).

Data information: Statistical significance was assessed by unpaired two-sided Student's *t*-test. As stated, data represented as mean $\pm$ SD or SEM, *$P$ < 0.05; **$P$ < 0.01; ***$P$ < 0.001.

(Lorenzi *et al*, 1986; Kaneto *et al*, 1996; Zhong *et al*, 2018). Subsequently, they were treated with 5 $\mu$M etoposide, a potent agent for inducing NHEJ-repair (Appendix Table S1). Irrespective of the type of reducing sugar used, all cells responded with an enhanced DNA damage response (DDR), as marked by the equi-distribution of $\gamma$H2AX foci in these cells. This shows that reducing sugars do not affect the initial DDR.

Moreover, cells cultured under low glucose conditions can repair their DNA-DSBs in a timely fashion (< 24 h), showing an efficient and functional DNA repair system under these conditions. However, cells cultured in high glucose, or even more pronounced ribose containing medium, were not able to repair their DSBs within 24 h (Fig 2A and B). This indicates a compromised DNA repair potential of these cells. This defective repair potential was further confirmed by immunoblotting (Fig EV2C). A similar but less pronounced response was observed, when another DNA-DSBs-inducing agent camptothecin (1 $\mu$M for 60 min; CPT) (Polo *et al*, 2012), which induces DNA-DSBs in the S-phase of the cell cycle, was used (Fig EV3A–F). Moreover, this decrease of DNA repair potential correlated with the reducing capacity of the carbohydrates used, with ribose and fructose having the most prominent effect. Thus, these data show that prolonged exposure to reducing sugars affects the DNA repair potential, especially the NHEJ-repair pathway, of these cells. Furthermore, to validate the adverse role of reducing sugars in DNA repair, laser-induced NHEJ-repair kinetics of hPARP and hXRCC4 was studied. When cells were grown in the presence of a high concentration of reducing sugars, the time course of repair (both recruitment and retention, marked by fluorescent red streak) was severely compromised (Fig 2C and Movie EV1). To further confirm that this impaired response is not associated with a single cell type alone, primary murine lung fibroblasts, podocytes, and human embryonic kidney (HEK-293), cells were also studied. Irrespective of the cell type used, reducing sugars affect the timely DNA repair (Appendix Fig S2A–D). Furthermore, the patho-physiological complications associated with elevated blood sugars develop over time. Thus, to simulate the pathological consequences of high blood sugar spikes in cultured cells in short time, we used high sugar concentrations. Moreover, considering the clinical scores of poorly managed diabetes in juvenile or a postprandial slot, the glucose levels in diabetics can be around 12–20 mM and the levels of other reducing sugars such as fructose or ribose can be around

(0.6–1.9 mM) or (~ 100 $\mu$M), respectively (Gross & Zollner, 1991; Sidhu *et al*, 2001; Clark *et al*, 2014; Laughlin, 2014; Chen *et al*, 2017; Wang *et al*, 2018). Thus, to simulate the pathologically relevant hyperglycemic conditions, the DNA repair studies were repeated with cells maintained under 17 mM glucose (for 5, 10, or 15 days). The control low glucose condition was the same as used previously (5.5 mM). Similar to the earlier observations, the cells grown under elevated levels of glucose cannot resolve the DSBs signaling within 24 h (Appendix Fig S3A), whereas the control cells maintained under low glucose conditions can repair its damage within this time. Moreover, the cellular ability to repair the DSBs was inversely linked to the duration for which these cells were maintained under high glucose (Appendix Fig S3A). Similarly, the addition of fructose and/or ribose to the high glucose further decreases the DNA repair capacity of these cells (Appendix Fig S3B). Consistent results were obtained in HEK-293 cells as well (Appendix Fig S3C and D). This cell-based study conclusively points toward a decreased DNA repair potential of cells maintained under hyperglycemic conditions via an unknown mechanism.

To further elucidate the effect of reducing sugars on DNA repair, an NHEJ-GFP reporter cell line (Seluanov *et al*, 2010a) was studied. The reducing potential of the carbohydrates correlates with a decrease of NHEJ-repair, with ribose being the most potent agent reducing DNA repair (Fig 2D). PARP and SIRTs are important DNA repair factors involved in NHEJ-repair. Their activity is under the regulation of nicotinamide cofactors (Canto *et al*, 2013; Li *et al*, 2017). Reducing sugars are known to affect the redox pool (NAD$^+$ and NADH) of cells (Lorenzi *et al*, 1986; Charron & Bonner-Weir, 1999). Therefore, the defective repair potential of cells cultured under high concentrations of reducing sugars might be related to the unwarranted perturbation of the metabolic balance affecting nicotinamide cofactors. When the ratio of oxidized to reduced nicotinamide adenine dinucleotide (NAD$^+$/NADH) was determined, increased concentrations of reducing sugars shifted the equilibrium toward a decreased NAD$^+$ and increasing NADH cofactor pool (Fig 2E). A depleted NAD$^+$ pool is known to play an important role in affecting the free availability of PARP for DNA repair (Charron & Bonner-Weir, 1999; Yan, 2014; Li *et al*, 2017). Therefore, immunoprecipitation of PARP was used to test if increased doses of reducing sugars affect the PARP-DBC1 interaction, known to prevent the participation of PARP in DNA-DSBs

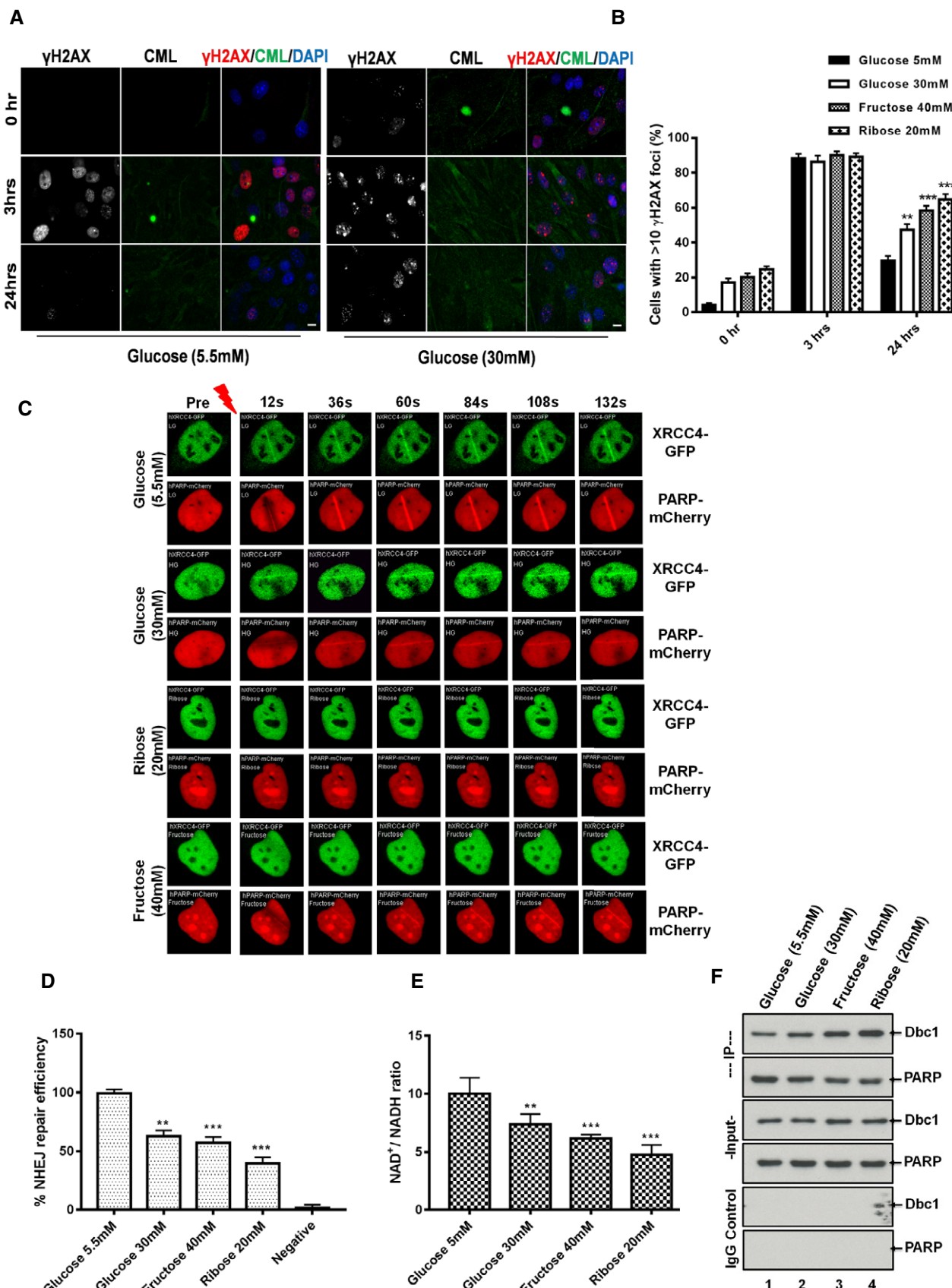

Figure 2.

**Figure 2.  Prolonged exposure of high concentration of reducing sugars affect the kinetics of DNA-DSBs repair.**

A  Immunofluorescence analysis of DSBs-associated foci as marked by γH2AX in lung adenocarcinoma (A549) cells, cultured either in low glucose (5.5 mM, left panel) or high glucose (30 mM, right panel) for 5 days, then treated with etoposide (5 μM for 60 min). The resolution of DNA-DSBs foci, as marked by γH2AX, was monitored over 24 h after drug treatment, CML (marked in green) served as induction control (scale 10 μm).

B  Mean percentage of DSBs-positive nuclei as evidenced by γH2AX positivity after pre-treatment of different reducing carbohydrates at 0, 3, or 24 h. More than 400 cells were analyzed for each bar (mean ± SD, **$P < 0.01$; ***$P < 0.001$).

C  Still images of A549 cells showing the live recruitment of hPARP-mCherry and hXRCC4-GFP, at the site of laser-induced DNA-DSBs from the cells pre-treated with the indicated reducing sugars as described in Methods.

D  Quantitative analysis of the efficiency of NHEJ-repair in a reporter cell line, as described in Methods. Cells were cultured in low glucose (5.5 mM) or high glucose (30 mM) for 5 days, fructose (30 mM), ribose (20 mM), for 3 days (graph bar 4). *I-SceI*, untransfected cells served as a negative control. Shown is the average from three independent experiments (mean ± SEM, **$P < 0.01$ ***$P < 0.001$).

E  Quantitative analysis of the NAD$^+$/NADH ratio in A549 cells, cultured in either low (5.5 mM, LG) or high glucose (30 mM, HG) for 5 days, and fructose (30 mM, FR) and ribose (20 mM, RI) for 3 days (graph bar 4). Quantification was performed after homogenization in extraction buffer (data represent mean ± SD, **$P < 0.01$; ***$P < 0.001$; $N = 3$).

F  A549 cells were cultured in either low (5.5 mM; lane 1) or high glucose (30 mM; lane 2) for 5 days, and fructose (30 mM; lane 3) and ribose (20 mM; lane 4) for 3 days. The cell extracts were then for immunoprecipitation by using anti-PARP, or a non-specific species control antibody. The PARP, or its interacting partner DBC1, was then detected using PARP, or a DBC1-specific antibody.

Data information: Statistical significance was assessed by unpaired two-sided Student's *t*-test. As stated, data represented as mean ± SD or SEM, **$P < 0.01$; ***$P < 0.001$.

repair. As expected, the PARP-DBC1 interaction is increased in the presence of 30 mM glucose compared to 5 mM glucose, but even more, by the other reducing sugars tested (Fig 2F and Appendix Fig S4A). Inclusion of NAD$^+$ (500 μM) in the wash buffer disrupts this interaction, confirming that the increased PARP-DBC1 interaction was indeed linked to the depletion of the NAD$^+$ pool (Appendix Fig S4B). Furthermore, NADH pre-treatment is known to shift the cellular NAD$^+$/NADH equilibrium toward NADH (Ma *et al*, 2011; Kembro *et al*, 2014), hence can be used to verify if hyperglycemia-associated impairment of DNA-DSBs repair is causatively linked to a change in the NAD$^+$/NADH pool. To assess this, A549 cells (maintained under 5.5 mM glucose) were pre-treated with NADH (100 or 250 μM for 24 h) and the DNA-DSBs repair kinetics was studied by using the DSBs signaling marker γH2AX. Notably, NADH treatment drastically reduced the DNA repair capacity of these cells, as did the hyperglycemic environment (Appendix Fig S4C and D). Thus, elevated levels of reducing carbohydrates, including glucose, affect genomic integrity in two ways. (i) Prolonged exposure of cells to hyperglycemia results in shifting the NAD$^+$/NADH equilibrium toward NADH. This cellular condition is also called as "reductive stress" (Yan, 2014; Luo *et al*, 2016). As under this condition, the amount of NADH produced exceeds the capacity of the electron transport chain and results in leakage of the electrons and an improper reduction of O$_2$ to the highly reactive superoxide anion (Fisher-Wellman & Neufer, 2012). Moreover, mitochondria react to this modified situation (reductive stress and over-nutritioned cell) by generating more and more H$_2$O$_2$. Cells respond to these situations by up-regulating the expression of antioxidant pathways (Nrf2 etc.), therefore leading to increased levels of GSH (Korge *et al*, 2015). This contributes further to the reductive stress. Thus, under this highly reduced environment, mitochondrial ROS production surpasses the scavenging capacity of the cell (Aon *et al*, 2010; Kembro *et al*, 2013; Cortassa *et al*, 2014), thus affecting the integrity of the genome and other biomolecules. (ii) Elevated levels of reducing carbohydrates affect the NHEJ DNA repair in part by keeping PARP bound to DBC1 and, thus, making it unavailable for the DNA-DSBs repair. This process depends directly on the redox balance because increasing NAD$^+$ can free the DBC1 bound PARP (Li *et al*, 2017). Taken together, these data conclude that a hyperglycemic

environment associated with defective DNA-DSBs repair is directly linked to the redox pool and acts independently from glycation-induced protein–protein cross-links.

Under several situations, like a compromised DNA repair system, severe or irreparable DNA damage reinforces the repair system toward a constitutive DDR signaling, as marked by a prolonged p53-dependent growth arrest and senescence. Considering this, cells maintained under hyperglycemic conditions show markedly elevated levels of unrepaired DSBs; therefore, it is important to distinguish whether this unrepaired DNA could impose an essentially irreversible growth arrest under hyperglycemic conditions and whether antioxidant therapies can be beneficial under these situations. To test this, A549 cells were pre-treated with the indicated reducing sugars for 5 days (with or without *N*-acetyl-cysteine; 2 mM; NAC). The DNA damage was induced by treating cells with etoposide (5 μM for 60 min). After the damage, cells were allowed to repair its DNA, and the persistent DNA damage signaling was studied as described earlier (Rodier *et al*, 2009). Here, it was observed that cells maintained under hyperglycemic conditions showed marked cellular senescence, as evidenced by senescence-associated β-galactosidase (SA-β-galactosidase (Debacq-Chainiaux *et al*, 2009), and addition of fructose and/or ribose under these conditions further enhanced it (Appendix Fig S5A). However, cells maintained under low glucose conditions show very little or no senescence, which can only be slightly induced by the presence of fructose and/or ribose. Remarkably, NAC treatment completely abolishes the cellular senescence in cells maintained under low glucose conditions but only slightly reduced it in the cells maintained under hyperglycemic conditions (Appendix Fig S5A). This observation was verified by quantitative analysis of the SASP marker IL-6 (Appendix Fig S5B). In addition, similar findings were obtained from renal cells (Appendix Fig S5C and D). Together, these data demonstrate that the hyperglycemic environment promotes a persistent DNA-DSBs signaling, which then modulates the cell cycle arrest, senescence, and SASP.

**Diabetes is associated with cellular senescence and persistent DNA damage signaling**

Several metabolic diseases are known to alter the molecular equilibrium and kinetics of DNA repair factors. In diabetes, inactive state

of PARP (particularly for DSBs repair) is associated with persistent DNA damage signaling (Charron & Bonner-Weir, 1999; Canto *et al*, 2013; Li *et al*, 2017). Thus, STZ-induced experimental diabetes model was used for studying the effects of hyperglycemia-induced metabolic stress in DNA repair *in vivo*. The lungs, as well as the kidneys, are prime targets of ROS-associated metabolic maladaptations (Tatsch *et al*, 2012; Giovannini *et al*, 2014; Shimizu *et al*, 2014; Bhatt *et al*, 2015; Burton & Faragher, 2018) (Clark *et al*, 2002; Tatsch *et al*, 2012; Bhatt *et al*, 2015; Wu *et al*, 2016; Simo *et al*, 2018). DNA-DSBs-associated pulmonary and renal consequences of diabetes were studied after STZ-induced diabetes (3 and 6 months; type 1 diabetes model). Type 1 diabetes was associated with increased activation of the DNA-DSBs pathway (Fig 3A and B, and Appendix Fig S6A and B) as well as oxidative stress markers in the lungs (Fig 3C). Like in lungs, diabetes-associated DNA damage as marked by DNA-DSBs signaling marker γH2AX (Appendix Fig S7A and B), as well as oxidative stress (Appendix Fig S7C and D), was significantly enhanced in diabetic kidneys as compared to the age-matched non-diabetic controls. This indicates that the studied patho-mechanism affects both organs similarly. Furthermore, persistent DNA damage signaling is known to be associated with cellular senescence and SASP as marked by inflammatory cytokines such as IL-6 and IL-8 (Rodier *et al*, 2009; Kumar *et al*, 2017). However, under certain situations, resident macrophages also show false positivity for SASP markers (Hall *et al*, 2016, 2017). Thus, to explicitly separate diabetes-induced DNA-DSBs-associated persistent DNA damage signaling from macrophages, only images with alveolar epithelial cells were used for quantification and presentation. Here, it was observed that the markers of persistent DNA damage signaling, such as IL-6 (Fig 3D, and Appendix Fig S7E) and β-gal-mediated cellular senescence (Fig 3E), were also increased in a type 1 diabetic mice model when compared to age-matched controls. In order to further prove that elevated blood glucose levels are causatively linked to persistent DNA damage signaling in the STZ-induced diabetic model, a type 2 diabetes ("*db/db*") model was studied. In this model, blood sugar levels start rising after 1.2–1.8 months of age, and after that, persistent DNA damage signaling gradually develops over time. In order to evaluate the DNA repair and persistent DNA damage signaling in this model, 4-month-old obese *db/db* mice, along with lean controls, were studied. In addition to type 1 diabetic model, *db/db* mice also showed elevated markers of DNA-DSBs signaling, as evidenced by γH2AX in both lung (Fig EV4A and B) and kidney (Fig EV4C and D). Furthermore,

similar to the STZ model, these DNA-DSBs were also associated with persistent DNA damage signaling, as evidenced by the SA-β-galactosidase, which was markedly enhanced in both, lung and kidney, of *db/db* as compared to lean controls (+/*db*)(Fig EV4E). Altogether, these data demonstrate that irrespective of the type of diabetes model used, elevated sugar coupled ROS is associated with compromised DNA-DSBs repair system and persistent DNA damage signaling, which might interfere with the normal physiology of these organs.

## Persistent DNA damage signaling is associated with pulmonary and renal fibrosis in diabetes

Persistent DNA damage signaling and SASP is known to induce organ fibrosis. To test whether diabetes-associated DNA-DSBs signaling is linked to organ fibrosis, the lung function of the type 1 diabetic mice was studied. Diabetes results in a significant decrease in lung function, mimicking the condition of restrictive lung disease (Fig 4A and B, and Appendix Fig S8A). To further clarify that the decreased lung function was associated with fibrosis, the lung sections of these mice were analyzed by Masson's trichrome staining, and it was confirmed that decreased lung function of the diabetic mice is associated with a marked accumulation of extracellular matrix, such as collagen (Fig 4C). This disturbed parenchymal texture of these lungs was also verified by H&E staining (Appendix Fig S8A). Similar to lungs, the hyperglycemia-associated matrix accumulation and altered tissue texture were also observed in the type 1 diabetic kidney (Appendix Fig S8B and C) as well as in type 2 diabetic lung and kidney (Appendix Fig S9A and B). Hence, irrespective of the type of diabetes model used, the persistent DNA-DSBs-associated signaling affects these two organs equally.

Moreover, to further elucidate the relevance of our murine model of diabetes, the relation of impaired DNA repair to organ fibrosis was studied in patients with diabetes (type 1 and type 2; Appendix Table S2). Here γH2AX positivity of mononuclear cells was used as a marker for ROS-mediated DNA damage signaling (Appendix Fig S10). γH2AX correlated positively with increasing albuminuria and negatively with decreasing diffusion lung capacity (DLco) (Appendix Fig S10). Diabetic patients also showed a marked decrease in total lung capacity (Fig 4D) and forced vital capacity (FVC) (Fig 4E). Thus, the murine and human disease correlate well with the models studied.

Furthermore, consistent with the data obtained using cultured cells, analysis of lungs from STZ mice also showed a significant

**Figure 3. Diabetes-associated persistent DNA damage signaling is associated with cellular senescence and SASP.**

A  Representative images of lungs for nuclei positive for the DNA-DSBs marker γH2AX, versus DAPI, in age-matched 3-, or 6-month control, versus STZ-induced diabetic mice, as determined by immunofluorescence analysis (scale 10 μm).

B  Mean percentage of nuclei positive for the DNA-DSBs marker γH2AX in lungs of age-matched control versus 6-month STZ-induced diabetic mice, as determined by immunofluorescence analysis of lungs (mean ± SEM, *P < 0.05; **P < 0.01; ***P < 0.001; N = 6).

C  Representative images of lungs from age-matched control versus 3- or 6-month STZ mice, stained for the peroxide linked oxidative stress marker DHR-123(in red), as determined by mean fluorescence intensity. Blue nuclear staining represents DAPI (scale 10 μm).

D  Quantitative analysis of respectively age-matched control versus 3- or 6-month STZ-induced diabetic lungs for persistent DNA damage signaling-associated inflammatory marker IL-6, as determined by mean fluorescence intensity of respective group lungs (mean ± SD; **P < 0.01, ***P < 0.001; N = 6).

E  Representative images of lungs from age-matched controls, versus 3- or 6-month STZ-induced diabetic mice. Sections stained for cellular senescence-associated β-galactosidase [β-Gal] as described in Methods and visualized by bright field and polarized light; the senescent areas are recognized by its bluish-green staining (scale 40 μm).

Data information: Statistical significance was assessed by unpaired two-sided Student's *t*-test. As stated, data represented as mean ± SD or SEM, *P < 0.05; **P < 0.01; ***P < 0.001.

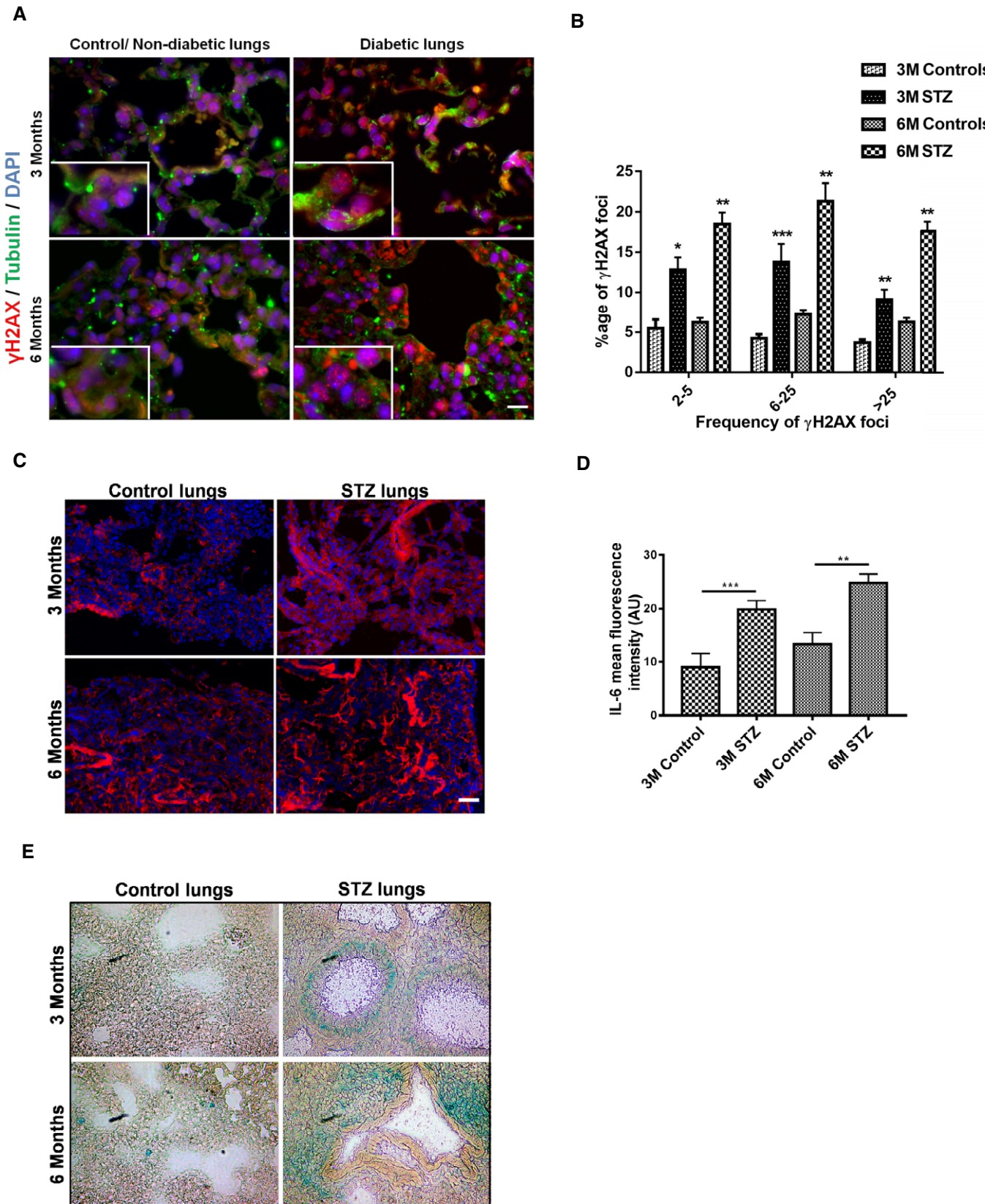

**Figure 3.**

decrease in NAD$^+$/NADH ratio (Fig 4F). A similar decrease in NAD$^+$/NADH ratio was also observed in age normalized patients with diabetes (Fig 4G). Thus, diabetes directly affects the non-homologous DNA repair potential of PARP. This renders cells to become vulnerable to the endogenous oxidative challenges and thereby leads to genomic instability. Lack of timely repair ultimately pre-disposes patients with diabetes toward persistent DNA damage signaling, senescence, inflammation, and organ fibrosis.

### Restoring DNA-DSBs repair ameliorates senescence, senescence-associated secretory phenotype as well as fibrosis

Nuclear RAGE plays a vital role in DNA repair (Kumar *et al*, 2017). Restoring nuclear RAGE reverses DNA damage, senescence, and fibrosis and improves the decreased lung function in a RAGE$^{-/-}$ mouse model of pulmonary fibrosis (Kumar *et al*, 2017). If other DNA-DSBs-associated fibrosis model such as diabetic mice also displays rescue by over-expression of nuclear RAGE. To evaluate this, diabetic mice were treated with constitutive and global gene expression serotypes of AAV [AAV2/8 (Franich *et al*, 2008; Li *et al*, 2010; Chung *et al*, 2011; Payne *et al*, 2016)] virions (Appendix Table S3) carrying a phosphomimetic RAGE mutant (RAGE$^{S376E–S389E}$), leading exclusively to nuclear RAGE expression (Fig EV5A and B), thus avoiding transmembrane (Kumar *et al*, 2017) or mitochondrial (Kang *et al*, 2011) RAGE signaling. A vector expressing RFP alone was used as a negative control. Furthermore, to understand the role of ATM signaling, a non-phosphorylatable mutant of RAGE (RAGE$^{S376A–S389A}$) was used. When the phospho-mimetic mutant (RAGE$^{S376E–S389E}$) was transduced in STZ mice, diabetic for 6 months, a drastic reduction of the DNA-DSBs-associated γH2AX foci was seen (Fig 5A and B). Specific intra-nuclear foci were seen in the lungs of diabetic animals (Fig 5A; panel 2, vector alone). The γH2AX foci were almost completely normalized in STZ animals treated with phosphomimetic RAGE (Fig 5A; panel 3), whereas the non-phosphorylatable RAGE has almost no effect on the quality and quantity of γH2AX foci (Fig 5A; panel 4). Similar data were obtained when pATM-positive foci (Fig EV5C and D) were studied. This was accompanied by a marked reduction in the positivity of senescence-associated β-galactosidase (Fig 5C), IL-6 (Fig 5D), and a decrease of extracellular matrix components (Fig 5E and F). More importantly, the reduction in DNA damage foci and

senescence was also accompanied by improved lung function (Figs 5G and EV5E). However, the non-phosphorylatable RAGE mutant (RAGE$^{S376A–S389A}$), or vector alone, had only a minor, or no effect (Fig 5G; red or green line, EV5E). These improved phenotypic changes were also confirmed by the quantitative mRNA analysis of the inflammatory cytokines from the respective group (Fig EV5F). Besides the lung, transduced diabetic kidney also showed that phosphomimetic RAGE reduces the γH2AX foci (Appendix Fig S11A and B), and the accumulation of ECM components (Appendix Fig S11C) as well as levels of inflammatory cytokines (Appendix Fig S11D). Further, as there are very limited numbers of biochemical parameters available to distinguish the progression or regression of renal fibrosis, the post-transduction-associated functional changes in the kidney, such as creatinine excretion and urine output, were studied. Mice treated with the phosphomimetic RAGE mutant (RAGE$^{S376E–S389E}$) showed significant improvements in both parameters (Appendix Fig S11E and F), while the non-phosphorylatable RAGE (RAGE$^{S376A–S389A}$) does not improve renal functions. Despite a striking effect on fibrosis, RAGE treatment did not reduce albuminuria, indicating that the vascular leakage leading to albuminuria and fibrosis are two distinct phenotypes of renal damage in diabetes. This observation was in agreement to the previous reports which showed that progression or regression of renal fibrosis is not related to albuminuria (Magalhaes *et al*, 2017). This further indicates that the patho-mechanisms leading to albuminuria and fibrosis are distinct. Thus, a phosphomimetic RAGE mutant, localized in the nucleus, reverses DNA damage and thereby senescence, the SASP, fibrosis and restores organ function, gives room for the speculation, that diabetes-associated fibrosis might at least in part be reversible.

## Discussion

From this study, a novel unifying hypothesis of diabetic complications emerges. In this model, the individual with diabetes, but free of complications, can timely repair metabolism-induced DNA damage. This avoids the persistent damage response and thus prevents the uncontrolled "Turn-on" of senescence induced paracrine action of inflammatory cytokines (Rodier *et al*, 2009; Kumar *et al*, 2017). However, in patients with diabetic complications, or with diabetes and radiation as part of cancer therapy, the capacity

---

**Figure 4. Persistent DNA damage response correlates well with the diabetes-associated fibrosis.**

A  Pressure–volume curves were determined using the FlexiVent system in age-matched control versus 3-month STZ-induced diabetic mice. The curves represent group averages (mean ± SD; *$P < 0.05$; $N = 5$).

B  Quantitative analysis of static pulmonary compliance in age-matched control versus 3-month STZ-induced diabetic mice, as described in Fig 3A. The curves represent group averages (mean ± SD; *$P < 0.05$, **$P < 0.01$; $N = 5$).

C  Representative images of lungs from age (3 or 6 months)-matched control versus STZ-induced diabetic mice, stained for Masson's trichrome stain, as described in Methods and visualized by bright field and polarized light, the accumulated ECM is recognized by its blue staining (scale 40 μm).

D  Percentage total lung capacity of controls versus patients with diabetes. The dotted line represents the cutoff decided as per the guidelines of the European Respiratory Society (ERS) and American Thoracic Society (ATS) (mean ± SD, *$P < 0.05$; triplicates from each $N^{control} = 44$, $N^{type\ 1} = 35$, $N^{type\ 2} = 110$).

E  Percentage forced vital capacity of lungs in controls versus patients with diabetes. The dotted line represents the cutoff decided as per the guidelines of ERS and ATS (mean ± SD; *$P < 0.05$; triplicates from each $N^{control} = 44$, $N^{type\ 1} = 35$, $N^{type\ 2} = 110$).

F  Quantitative analysis of the NAD$^+$/NADH ratio in lungs harvested from age-matched 3-, or 6-month control, or STZ-induced diabetic mice, after homogenization in extraction buffer (mean ± SD; **$P < 0.01$, ***$P < 0.001$; $N = 5$).

G  Quantitative analysis of the NAD$^+$/NADH ratio determined in serum from patients with diabetes, as described in Methods ($N = 3$; mean ± SD, *$P < 0.01$, ***$P < 0.001$).

Data information: Statistical significance was assessed by unpaired two-sided Student's *t*-test. As stated, data represented as mean ± SD, *$P < 0.05$; **$P < 0.01$; ***$P < 0.001$. One-way ANOVA was used for panel (D and E).

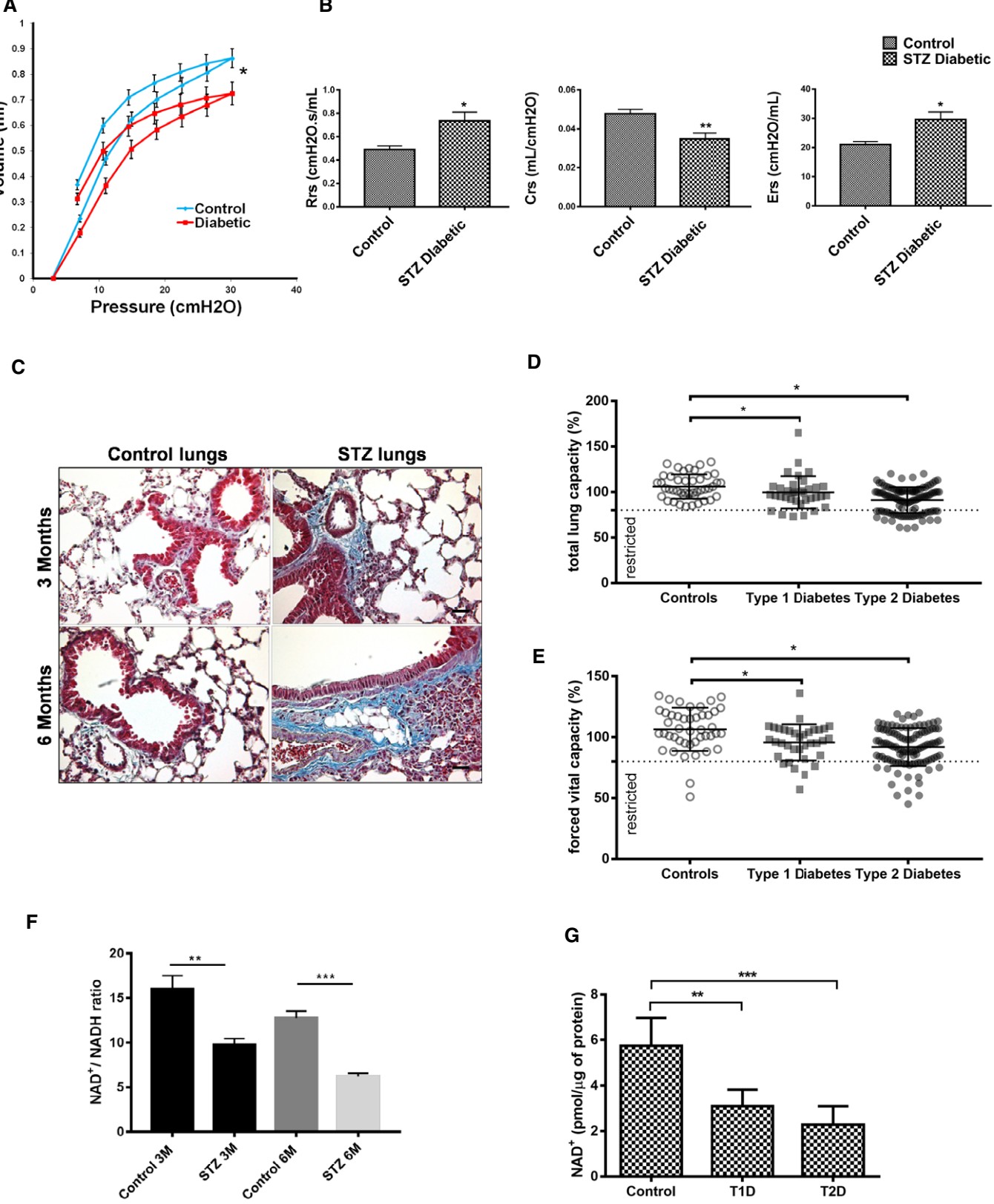

Figure 4.

to timely adapt to metabolic reprogramming and induction of DNA repair is lost (Charron & Bonner-Weir, 1999; Canto et al, 2013). Loss of the DNA repair potential results in persistent DNA damage signaling, senescence, SASP, fibrosis, and organ failure (Charron & Bonner-Weir, 1999; Debacq-Chainiaux et al, 2005; Rodier et al, 2009; Kumar et al, 2017; Kim et al, 2019). These observations are markedly supported by the clinical data and the study of the NAD$^+$ pool presented here. Furthermore, this might explain the increased radiation sensitivity of the microvasculature, lung, and skin of patients with diabetes (Clark et al, 2002).

In newborns, NAD$^+$-driven DNA repair correlates well with elevated oxidative stress and DNA damage. We hypothesized that a similar mechanism might help to override metabolic disease-induced DNA damage in actual diabetic mice too. By using phosphomimetic nuclear RAGE as a tool for improving DNA repair, the potential to reduce diabetic complications was shown in experimental diabetes. Furthermore, the RAGE transduction was initiated after the progression of fibrosis, thus demonstrating that nuclear RAGE-assisted DNA repair reduces fibrosis. Since the levels of γH2AX in *ex vivo* isolated mononuclear cells of diabetic patients correlate significantly with pulmonary dysfunction, as well as, with albuminuria (Kopf & Nawroth, 2018; Kopf et al, 2018), therefore the activation of the innate immune response and inflammation seen in diabetic patients are causally connected to DDR (Blasiak et al, 2004; Giovannini et al, 2014; Tornovsky-Babeay et al, 2014; Bhatt et al, 2015). The lack of an association with diabetic retinopathy suggests an organ-specific role of DNA damage and the repair pathway studied here.

Moreover, in the mouse model presented, a decrease in the NAD$^+$/NADH ratio plays a central role in impairing the DNA-DSBs repair, since addition of NAD$^+$ is capable of improving DNA repair (Li et al, 2017). The data presented indicate that ROS plays a central role in hyperglycemia-associated DNA damage, as well as in impairing DNA repair, but it does not exclude other form of DNA damages that might also occur. Furthermore, specific tests for different components of the redox balance are subject to future studies.

This study focused on ROS-mediated DNA double-strand breaks (DSBs) only. It remains unknown whether excessive ROS formation underlies these abnormalities, or whether a decreased antioxidant defense or the absence of timely DNA repair contributes to persistent DNA damage signaling-associated fibrosis. Furthermore, hyperglycemia-associated reductive stress is causatively linked to increased levels of NADH (Yan, 2014; Luo et al, 2016). Therefore, it interferes with the normal functioning of mitochondria, thus generates H$_2$O$_2$, as well as highly reactive superoxide anions, which further compromises the cellular physiology (Fisher-Wellman & Neufer, 2012). As a feedback control, cells up-regulate the expression of nuclear factor erythroid-related factor-2 (Nrf-2) to pump the reduced glutathione (GSH) into the system (Korge et al, 2015). This further contributes to reductive stress. Thus, increased ROS then targets cellular biomolecules such as DNA, RNA, and proteins (Lorenzi et al, 1986). ROS is also being generated as a consequence of the DNA damage signaling itself (Rodier et al, 2009). An additional explanation for the increased ROS formation stems from the experiments using different reducing carbohydrates. The stronger the reducing capacity of these carbohydrates, the more pronounced the ROS formation and DNA damage, and the reduction of DNA repair is seen. Moreover, under hyperglycemic conditions, sugars

are fluxed into the polyol pathway to generate sorbitol, which later is converted to fructose and further depletes the levels of NAD$^+$ (Brownlee, 2005; Yao & Brownlee, 2010). However, which of these mechanistic cross-talks is the most important for maladaptive DNA repair to metabolic stress in diabetes remains yet to be studied. Furthermore, future studies need to address the question of whether glycolysis and the Maillard reaction are casually involved in impaired DNA repair.

The interference of high glucose to the liberation of PARP from the inhibitory DBC1-PARP complex is one of the several pathways by which DNA repair might be impaired in diabetes (Charron & Bonner-Weir, 1999; Canto et al, 2013; Li et al, 2017). Recently, it has been shown that ATM-mediated phosphorylation of RAGE is central for DNA-DSBs repair, as the absence of timely DNA repair or the absence of RAGE (RAGE$^{-/-}$ mice model) is causatively linked to tissue fibrosis and cancer (Kumar et al, 2017). Reducing sugars such as glucose, as well as ROS, have been shown to interfere with the enzyme activity required for DNA repair, leading to inefficient DNA repair further downstream of RAGE and insufficient formation of MRE11-mediated resection complex at the site of DNA-DSBs. Furthermore, other pathways downstream of RAGE and the MRN complex are relevant for DNA repair (Canto et al, 2013; Charron & Bonner-Weir, 1999; Polo et al, 2012; Sartori et al, 2007). Some of these have been shown to be inhibited in patients with diabetes (Charron & Bonner-Weir, 1999; Canto et al, 2013). Therefore, it is likely that the depletion of NAD$^+$ reserve results in reduced NHEJ-repair, a pathway accounting for more than 75% of the DNA-DSBs repair in differentiated tissues (Mao et al, 2008); therefore, the persistent DNA-DSB signaling prevails in diabetes. Nevertheless, over-expression of a phosphomimetic mutant of RAGE seems to override the NHEJ defects by inducing homology-directed repair, which not only deactivates persistent DNA damage signaling but also stimulates tissue regeneration by yet unidentified mechanisms. If this holds in other diabetes models and also later stages of fibrosis remain to be studied.

Moreover, in addition to the findings described here, several independent groups also presented a similar observations in non-diabetic contexts, showing that persistent DNA damage signaling associated with fibrosis can be reduced by turning-off the persistent DNA-DSBs signaling (Armanios et al, 2007; Armanios, 2012; Cheresh et al, 2013; Chilosi et al, 2013; Liu et al, 2013; Svegliati et al, 2014; Povedano et al, 2015). Currently, there is no specific therapy aiming to reduce already existing diabetic complications. Cell surface RAGE and its ligand interactions are involved in the cellular cascade that leads to an inflammatory phenotype *in vitro* and *in vivo* (Bierhaus et al, 2001; Sparvero et al, 2009; Putranto et al, 2013; Sorci et al, 2013; Cai et al, 2016). Thus, this postnatal decrease of RAGE expression might be protective against activation of the innate immune system by preventing unwarranted receptor/ligand interaction, operative in several diseases associated with activation of RAGE-triggered inflammation. RAGE, therefore, has a dual function: as a cell surface receptor, it is part of pro-inflammatory response (Kokkola et al, 2005; Bucciarelli et al, 2006), whereas, in the nucleus, it is part of the DNA repair machinery (Kumar et al, 2017). Nuclear RAGE, along with the MRN complex, participates in DNA end resection. Accurate end-resection alienates ATM-mediated signaling cascade and activates ATR and checkpoint-1 (CHK-1) signaling (Cimprich & Cortez, 2008; Shiotani & Zou, 2009; Awasthi et al, 2015). Thus, loss of

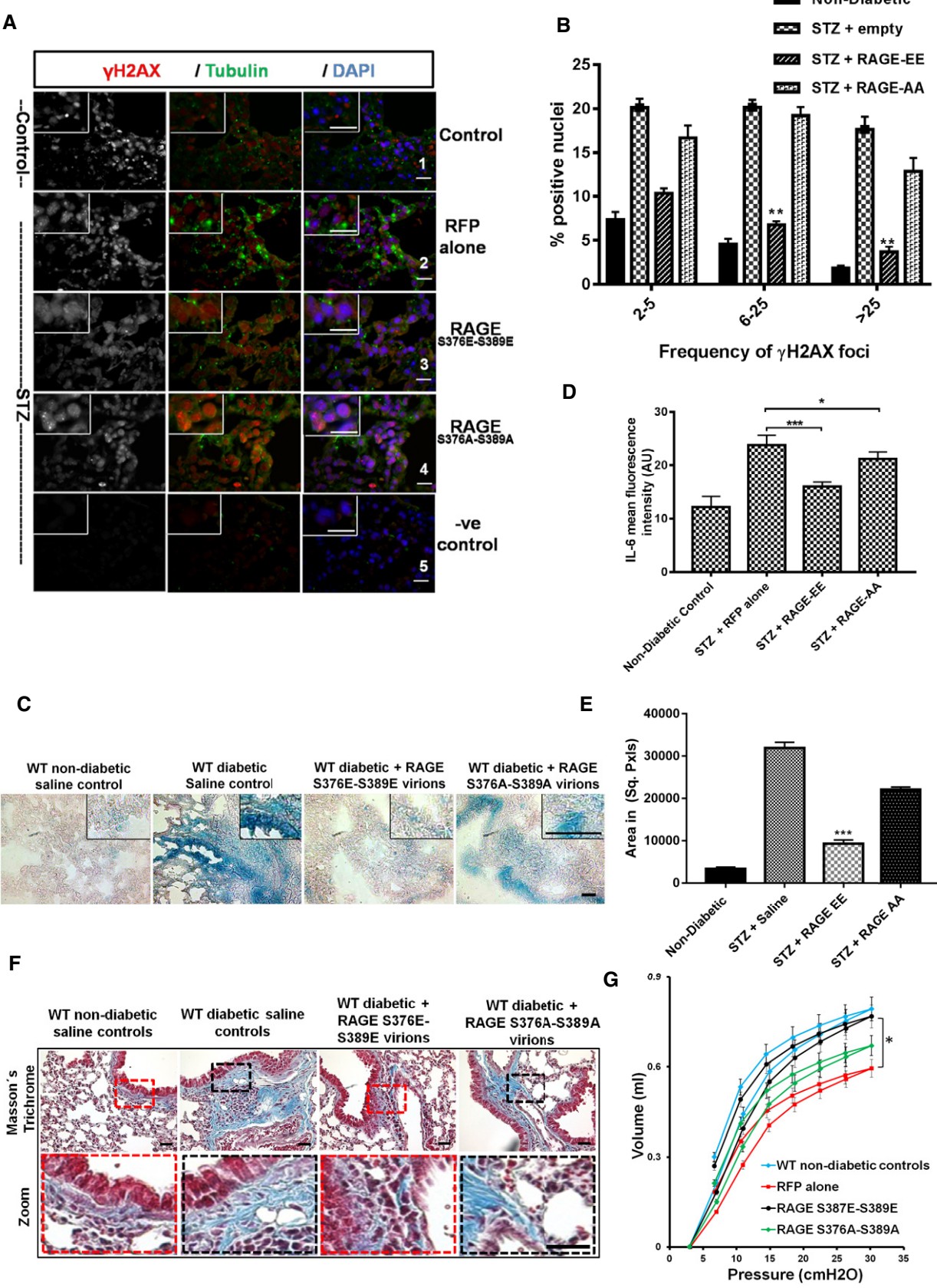

Figure 5.

◀

**Figure 5. Diabetes-associated persistent DNA damage signaling can be reversed by complimenting an alternative mode of DNA repair.**

A  Representative images of γH2AX-positive nuclei in lungs of 6-month STZ-induced diabetic mice, transduced with respective RAGE virions as described in Methods. The lungs were harvested 6 weeks after viral transduction. The empty vector served as control. Red represents γH2AX foci; green cytoplasmic staining represents α-tubulin; blue nuclear staining represents DAPI (scale 10 μm). No primary antibody control served as a negative control (from control lung) of the staining.

B  Mean percentage of nuclei positive for the DNA-DSBs marker γH2AX in transduced lungs of 6-month STZ-diabetic mice as described in (A) (mean ± SEM; **$P < 0.01$; $N = 8$).

C  Representative images of lungs from 6-month STZ-induced diabetic mice, transduced with respective RAGE virions as described in Methods. Lungs were stained for cellular senescence-associated β-galactosidase as described in (A) and visualized by bright field and polarized light, where the accumulated senescent areas are recognized by its blue staining (scale 40 μm).

D  Quantitative analysis of persistent DNA damage-associated inflammatory marker IL-6, in transduced lungs from 6-month STZ-diabetic mice, as determined by mean fluorescence intensity of respective group lungs (mean ± SD, *$P < 0.05$, ***$P < 0.001$; $N = 8$).

E  Quantitative analysis of Masson's trichrome stain for lungs of 6-month STZ-induced diabetic mice with respective RAGE virions as described in (F) (mean ± SD, ***$P < 0.001$; $N = 8$).

F  Representative images of Masson's trichrome stain for lungs transduced with the respective RAGE expressing virions as described in (A) and visualized by bright field and polarized light, where the accumulated ECM areas are recognized by its blue staining (scale 40 μm). The dotted lines represent the respective zoomed window.

G  Pressure–volume curves of diabetic and non-diabetic mice were determined using the FlexiVent system 6 weeks after transduction using respective the RAGE virions, as described in (A). The curves represent group averages ($N = 8$). The lungs studied in (A–D) are identical (mean ± SD; vector alone versus RAGE-EE; *$P < 0.05$; $N = 8$).

Data information: Statistical significance was assessed by unpaired two-sided Student's *t*-test. As stated, data represented as mean ± SD, *$P < 0.05$; **$P < 0.01$; ***$P < 0.001$.

constitutive RAGE expression might be at the cost of impaired adaptation to metabolic stress that indeed challenges the integrity of the double helix. This may lead to impaired DNA repair in situations such as aging-associated diabetes, where increased ROS and changes of carbohydrate metabolism induce severe stress on the cellular genome. This indicates that the metabolite-driven impairment of DNA repair is the main culprit for diabetic complications (Bhatt *et al*, 2015). This points to a complex architecture of DNA repair mechanisms, in which different repair pathways overlap and can be activated by compensatory mechanisms if one path to DNA repair is hampered. This is corroborated by the survival of RAGE$^{-/-}$ pups and some patients with a long duration of diabetes without complications (Constien *et al*, 2001; Tornovsky-Babeay *et al*, 2014; Bhatt *et al*, 2015). Though RAGE$^{-/-}$ mice survive well and are fertile, a later increase in pulmonary fibrosis and cancer has been noted (Constien *et al*, 2001; Kumar *et al*, 2017). Therefore, the dramatic effect of phosphomimetic RAGE in fibrosis points to the pharmacologically strong action of RAGE, but suggests that RAGE is not the only DNA repair controlling protein. Further experiments are needed to determine how long the effects of phosphomimetic RAGE sustain DNA damage and fibrosis, and whether this effect can be seen in later and severe stages of diabetes.

Elevated glucose and HbA1c are clinical parameters used for diagnosing diabetes. But both are quite poor predictors of diabetic complications (Radin, 2014). In type 1 diabetes, the diabetes duration plus HbA1c explain together only 11% of diabetic retinopathy (Frank, 2015). Furthermore, in the landmark diabetes intervention studies UKPDS and DCCT, the individual decrease of HbA1c neither identified the patients protected from complications, nor did it predict remission of complications. In addition, the absolute risk reduction by lowering glucose is for most complications below 5% in the DCCT and even lower in the type 2 diabetes studies (Kopf & Nawroth, 2018; Kopf *et al*, 2018). A recent meta-analysis of the absolute risk reduction of nephropathy by glucose control in type 2 diabetes revealed only a 0.4% reduction in absolute risk for nephropathy (Kopf & Nawroth, 2018; Kopf *et al*, 2018). Thus, it implies that mechanisms other than elevated plasma glucose alone must be operative in mediating diabetic complications in diabetes type 1 and type 2.

The hypothesis presented here assumes a two-hit model, in which complications only develop if the intact DNA repair system no longer compensates the metabolic stress-mediated DNA damage. This strongly implies novel therapeutic approaches dealing with the loss of defense pathways, responsible for the development of complications. One example might be the phosphomimetic RAGE, due to its strong effects on DNA repair, senescence, SASP, and tissue fibrosis in experimental diabetes. Since similar findings pertaining to DNA damage, altered NAD$^+$/NADH ratio, senescence, and fibrosis are also observed in type 1 and type 2 diabetic patients, a more general concept of diabetes and its complications, including the enhanced radiation sensitivity and fibrosis formation of skin, lung, and microvessels in patients with diabetes emerges.

In this concept, acquired disorders of metabolism may trigger, by various pathways, even independent of the plasma glucose concentration, toxic metabolite formation. These mediate not only DNA damage, but also by impairing DNA repair trigger a sequence of events leading to senescence, inflammation, tissue fibrosis, and finally to organ dysfunction. Therefore, metabolite-induced DNA damage, DDR, and persistent DNA damage signaling are common soil for several complications of diabetes. Recognition of this common soil may lead to novel therapies, including phosphomimetic RAGE, aiming not only to prevent but rather to reduce diabetes-induced organ fibrosis and dysfunction.

## Methods

Details of plasmid, antibodies, DNA damage agent/treatment, information on DNA oligo(s) used are described in Appendix Tables. All experiments or assays were at least repeated thrice with a different batch of study models or unless stated specifically.

### Mouse models

Wild-type *C57BL/6*, *+/db*, and *db/db* (*C57BL/KSJRj-db*) mice were obtained from Janvier, France. In diabetic mice subgroup, persistent hyperglycemia was induced by i.p. administration of 60 mg/kg STZ for five consecutive days in 8-week-old mice. Age-matched control

mice received 100 μl PBS (1×) intraperitoneally for five consecutive days.

The STZ mice subgroup mice were considered diabetic if blood glucose levels were above 300 mg/dl 16 days after the last STZ injection. Blood glucose levels were determined from blood samples taken from the tail vein by using ACCU-CHEK glucose strips. In the first 3 weeks after the onset of diabetes, blood glucose values were measured three times per week and after that once per week. Mice with blood glucose levels above 500 mg/dl received 1–2 U of insulin (Lantus) to avoid excessive and potentially lethal hyperglycemia. Thus, the cohort was maintained somewhere between 350 and 450 mg/dl. Blood and tissue samples were obtained after 3 or (short-term model) or 6 months (long-term model) of persistent hyperglycemia in diabetic mice. Age-matched saline littermates served as controls. The procedure of the experiments was approved by the animal care and use committees at the Regierungs-spräsidiumTübingen and Karlsruhe, Germany.

## Ethics statement and tissue samples

All experiments were conducted in accordance with the Declaration of Helsinki and the International Ethical Guidelines for Biomedical Research Involving Human Subjects. The study was approved by the local ethics committee (ethics board approval S-206/2005, S270/2018, and S-284/2018). Pseudonymised archival tissue samples were retrieved from the tissue bank of the National Center for Tumor diseases (Heidelberg, Germany). Formalin-fixed and paraffin-embedded tissue samples were cut into 3-μm-thick sections and put on glass slides.

## Cell culture, DNA damage, and transfection

Lung fibroblasts, podocytes, HEK, or A549cells were grown in S-Syn-medium [DMEM (RPMI for Podocytes) + 10% FCS, 1% pen-strep, 1% glutamine) + 2 mM thymidine for 24 h (first block)]. After first thymidine block, cells were washed twice with PBS (1×) and grown in normal fibroblast growth medium (10% FCS, 1% pen-strep, 1% glutamine) for 3 h to release cells, after which the medium was changed to DMEM (10% FCS, 1% pen-strep, 1% gluta-mine), 100 ng/ml nocodazole was added to the cells for 12 h (second block). After the second block, thymidine was removed by washing with PBS and cells were released by adding fresh Phenol red free medium (10% FCS, 1% pen-strep, 1% glutamine) along with 10 μM BrdU to pre-sensitized for 24 h and then these S-phase cells were used in DNA damage treatment. The G-phase synchronization was induced by serum starvation of cells by growing them (for 60–72 h) in above-described growth medium, but supplemented with 0.1% serum. The isolation of primary murine lungs fibroblasts from murine lungs was performed as described earlier (Seluanov et al, 2010b). The drug treatment was performed as described in Appendix Table S1. The laser-induced DNA damage or live micro-scopy A549 cells was performed by pre-treating them with the reducing sugars as indicated (3 days, 2% FCS). In the last 24 h, cells were transfected with hXRCC4-GFP and PARP-mCherry constructs and continued the cultivation as before. The laser settings used were same as described previously (Kumar et al, 2017)

The plasmid DNA transfections in primary cells and A549 were performed either using the Neon (Invitrogen, GmbH) transfection system or Turbofect transfection reagent (Thermo GmbH). HEK 293 cells were transfected using the standard $CaCl_2$ method of virus production. Transfected cells were analyzed after 24–48 h of transfection.

## Cell lysis and immunoprecipitation

Total cell extracts or tissue extracts (liquid $N_2$ grinded powder of respective tissue) were obtained by re-suspending them in 20 mM Tris–Cl pH 7.5, 40 mM NaCl, 2 mM $MgCl_2$, 0.5% NP-40, 50 U/ml benzonase, supplemented with protease and phosphatase inhibitors, and after 15 min of incubation on ice, the NaCl concentration was adjusted to 150/450 mM and then it was further incubated for 15 more minutes. The lysate was then centrifuged at 14 K for 15 min at 4°C, and at least 0.5–1.0 mg proteins were used per immune precipitation in IP buffer (25 mM Tris–Cl (pH 7.5), 150 mM NaCl, 1.5 mM DTT, 10% glycerol, 0.5% NP-40) supplemented with protease and phosphatase inhibitors. Endogenous proteins were captured onto protein A/G- magnetic beads. In addition, in some immunoprecipitation assays, 500 μM $NAD^+$ was used for additional column washes. Proteins were resolved by 4–20% SDS–PAGE (Mini-PROTEAN TGX Bio-Rad), transferred onto nitrocellulose (Pro-tran), and probed using the appropriate antibodies described in Appendix Table S4.

## ELISA

The 2–3 day before functional studies or tissue preparation was done, individual mice were placed in metabolic cages and urine samples were collected and quantified. We determined urine crea-tinine using creatinine colorimetric/fluorometric assay kit (Biovi-sion; K625-100) according to the manufacturer's instructions. Similarly, the IL-6 levels in the respective/indicated samples were quantified using IL-6 quantikine ELISA kit (R&D; M6000B) accord-ing to the manufacturer's instructions.

## Quantification of mitochondrial copy number

For mtDNA quantification, DNA was extracted from postnatal lungs with Proteinase K digestion and subsequent organic extraction. Mitochondrial DNA (mtDNA) was quantified with quantitative PCR with primers shown in Appendix Table S5. The relative mtDNA copy number was calculated from the ratio of mtDNA copies to nuclear DNA (nucDNA) copies. The relative fold change was then calculated based on the ΔΔCt method.

## RT PCR

The total RNA was extracted from lung or renal tissues of the indicated groups using the RNeasy Mini kit (Qiagen; 74104) according to the manufacturer's instructions and then reverse-transcribed into cDNA using high-capacity cDNA reverse tran-scription kit (ABI; 4368814) and subjected to quantitative PCR using SYBR green supermix (KAPA SYBR fast mix) and the primer described in Appendix Table S5. Each sample was tested in triplicate. The qPCR data were analyzed using the comparative threshold cycle (Ct) method. GADPH was used as an internal control.

**Trichrome Masson and senescence staining**

Staining was performed on lung sections after deparaffinizing them in xylol and rehydrating them through a series of 100% ethanol, 95% ethanol, 70% ethanol then washed extensively with water and then incubated with Weigert's iron hematoxylin solution for 10 min and then, washed extensively with distilled water and incubated with Biebrich scarlet-acid fuchsin solution for 10–15 min. After incubation, the slides were washed extensively and differentiated in phosphomolybdic–phosphotungstic acid solution for 10–15 min and transfer them to aniline blue solution and stain for 5–10 min. Rinse briefly in distilled water and differentiate in 1% acetic acid solution for 2–5 min. After this step, slides were washed again with distilled water and dehydrated immediately with 95% ethyl alcohol, absolute ethyl alcohol, and xylene. The slides were then mounted and analyzed.

**Senescence**

SA-β-gal positivity of cells/tissue areas was tested as described earlier (van der Loo *et al*, 1998; Debacq-Chainiaux *et al*, 2009); in brief, the described specimens were fixed in 2% formaldehyde/0.2% glutaraldehyde/PBS for 5 min. Slides were then washed and incubated with 5-bromo-4-chloro-3-inolyl-β-D-galactoside in *N*, *N'*-dimethylformamide (20 mg/ml), 40 mM citric acid/sodium phosphate, pH 6.0, 5 mM potassium ferrocyanide, 5 mM potassium ferricyanide, 150 mM NaCl, and 2 mM $MgCl_2$ and incubated at 37°C for 24 h. After incubation, cells/tissue sections were washed with PBS, mounted, and imaged using an Olympus inverted microscope.

**H&E staining**

De-paraffinized sections were used for hematoxylin–eosin staining; the sections were stained with hematoxylin about 10 min (30°C), water rinsed for 15 min, and then differentiation in acid solution by incubating them for 5–30 s until the slice get red, then rinse water for about several min to the section of the eye can be seen blue. These sections were then placed into 75%, 95%, 100%, l00% ethanol solution for 5 min each, and then, eosin dye staining was performed for about 2 min. The eosin-stained sections were then sequentially dehydrated by for 5 min each and then placed into xylene I solution and xylene II solution each for 5 min. The slides were then mounted in the mounting medium and then dried overnight before analyzing them under the microscope.

**Virus production**

The production of recombinant AAV virions in HEK293 cells was performed as described earlier (Lu *et al*, 2015). Cells were transfected with three plasmids for each AAV virus type to be packaged (Appendix Table S3). The triple transfection of HEK-293T cells was set up as follows: for each confluent T150 flask, 12.5 μg of AAV backbone plasmid, 25 μg pDP2 helper plasmid, and 12 μg capsid plasmid were added to 2.4 ml of sterile water in a 15-ml Falcon tube and then 330 μl of 2.5 M $CaCl_2$ was added to the mixture. Transfected cells were incubated at 37°C/5% $CO_2$. 16 h post-transfection, media was removed and replaced with fresh complete DMEM. After 96 h of transfection, packaging cells were lysed in packing lysis buffer (50 mM Tris, 150 mM NaCl at pH 8.4). Virions were purified and concentrated using an iodixanol gradient and concentrated using the Vivaspin centrifugal concentrator (50-KDa cutoff).

**Lung function**

*Murine*

To evaluate lung mechanics, invasive lung function analysis was performed as described earlier (Wielputz *et al*, 2011), In brief mice were anesthetized with sodium pentobarbital (80 mg/kg), tracheostomized, and placed on a small animal ventilator (FlexiVent system, SCIREQ, Montreal, QC, Canada). To prevent spontaneous breathing, mice were then paralyzed with pancuronium bromide (0.5 mg/kg) and ventilated with a tidal volume of 10 ml/kg at a frequency of (150 breaths/min) and a positive end-expiratory pressure of 3 cm $H_2O$ to prevent alveolar collapse. Pressure–volume curves with stepwise increasing pressure (PVs-P) were consecutively measured. All perturbations were performed until three acceptable measurements were achieved.

*Human*

Spirometry, body plethysmography, and carbon monoxide-based diffusion capacity measurements were performed, using the body plethysmograph PowerCube Body+ by Ganshorn Medizin Electronic (Ganshorn Medizin Electronic GmbH, Niederlauer, Germany). Lung function testing was performed by specialized trained technicians according to the guidelines and reference values of the American Thoracic Society (ATS) and European Respiratory Society (ERS) study as previously described (Kopf & Nawroth, 2018; Kopf *et al*, 2018). DLco-SB was determined using a standard measurement gas containing 21% oxygen, 10% helium, and 0.3% carbon monoxide. All measured parameters were adjusted to age, gender, and BMI. Diffusion measurements were adjusted to the current hemoglobin value. FVC, single breath diffusion capacity of the lung with carbon monoxide ($DL_{CO}$), and body plethysmographic measurement of total lung capacity (TLC) were given in % predicted and were used for analysis.

**Chest computed tomography and 6-min walking test**

Participants with increased breathlessness, restrictive lung function, and inexplicable decreased SB-DLco < 60% were asked to undergo multidetector computed tomography (CT) and a 6-min walking test (6-MWT) on the same day. Separate written informed consent was obtained for these procedures. Non-enhanced CT (iCT 256, Philips Medical Systems, the Netherlands) was acquired in full inspiratory breath-hold and supine position and reconstructed in a lung kernel with overlapping slices of 1.5 mm thickness. Two dedicated chest radiologists interpreted images according to ATS/ERS criteria for the diagnosis of pulmonary fibrosis (Kopf & Nawroth, 2018; Kopf *et al*, 2018).

For the 6-MWT, participants were instructed to walk a 300-m corridor forward and backward at their speed to get as much distance as possible within 6 min (Kopf & Nawroth, 2018; Kopf *et al*, 2018). The distance was recorded by a distance measuring wheel (Rolson 50799 by Rolson Tools Ltd., Reading, UK).

## Cellular immunofluorescence

Cells grown on poly-L-lysine coated coverslips (Thermo) or glass-bottom dishes(Ibidi) were fixed with 4% paraformaldehyde for 15 min at 4°C and permeabilized with 0.3% Triton X-100 in PBS for 5 min at room temperature. Cells undergoing drug treatment were processed differently, such that after drug treatment or laser-induced DNA damage, cells were pre-extracted with CSK buffer (10 mM PIPES (pH 7.0), 100 mM NaCl, 300 mM sucrose, 3 mM $MgCl_2$, and 0.01% Triton X-100) for 1 min at room temperature. Samples were then blocked in 5% bovine serum albumin and immune-stained using indicated primary antibodies (listed in Appendix Table S4) and secondary antibodies. Species adsorbed AlexaFluor 488/555/647 secondary antibodies were purchased from Abcam. Fluorescent images of infected and control cells were captured with a CCD camera connected to an inverted fluorescence microscope (Cell Observer, Carl Zeiss, GmbH, Göttingen, Germany). Samples were scanned using an ×63 oil objective. Images were further processed using ImageJ (Fiji) and Photoshop CS5 (Adobe).

## Tissue sections immunofluorescence

Paraffin-embedded tissue sections were de-paraffinized using a series of washes with xylol (10 min × 2 times), isopropanol (5 min × 1 times), 96% ethanol (5 min × 1 times), 85% (5 min × 1 times), 70% (5 min × 1 times), and then in aquadest (5 min × 1 times). Antigen retrieval was performed by incubating sections in retrieval buffer-A (10 mM Tris–Cl pH-9.0, 1 mM EDTA, 0.05% Tween-20) for 20 min at RT. Sections were then extensively washed with water and permeabilized using 0.1% aqueous solution of saponin for 30 min at room temperature. These permeabilized sections were then washed extensively with Tris-buffered saline/ 0.2% Triton X-100 (TBS-T) for 5 min × 3 times. These prepared sections were then blocked with antibody dilution/incubation buffer (10% goat serum in TBS-0.2% Triton X-100) for 45 min at room temperature. After incubation, the respective antibodies were diluted in the antibody dilution/incubation buffer and incubated at 4°Cfor 8/10 h. The sections were then washed with TBS-T (10 min; thrice). Fluorochrome-conjugated and species adsorbed respected secondary antibodies were then used to detect the signal. The negative control staining was performed simultaneously with the test marker staining; here, we used alone antibody dilution/incubation buffer instead of primary antibody specific to the DNA-DSBs marker presented in the corresponding panel. The rest of the steps like incubating with fluorochrome-conjugated secondary antibodies were performed the same way for all the staining.

The imaging of these stained sections was performed as described for the cellular immunofluorescence staining.

## NHEJ-GFP reporter assay

To study the NHEJ-repair efficiency, NHEJ-GFP reporter cells were used as previously described (Seluanov *et al*, 2010a). In short, these cells were transfected with I-*SceI* plasmid (5 μg) and pDsRed-N1 (0.1 μg; normalization control). pDsRed-N1 plasmid alone transfected cells served as −ve control (−ve control) of this assay. Under basal conditions, the cells were grown under 5 mM glucose conditions, whereas the reducing sugar pre-treatment (glucose, 30 mM; fructose,

40 mM; or ribose, 20 mM) was given for indicated days. After 72 h of transfection, the efficiency of NHEJ-repair was determined by the percentage of GFP$^+$ cells against the normalization control. Fluorescent-activated cell sorter (FACS) was used to sort cells expressing GFP positivity and RFP from the NHEJ-reporter cell line. The gating was done using untransfected GFP or RFP alone transfected cells.

## Reactive oxygen species determination

Dihydrorhodamine-123 (Life Technologies, D-23806) was reconstituted in DMSO. Freshly made cryosections of unfixed postnatal lungs were incubated in 10 mM dihydrorhodamine-123 in PBS for 20 min in the dark and then mounted with DAPI containing Vectashield mounting medium.

## NAD$^+$/NADH quantification

NAD$^+$/NADH ratio was quantified by using NAD/NADH Quantification Kit (Biovision; Catalog number. K377A) by following the manufacturer's instructions and normalized to total protein content. To eliminate the NAD or NADH consuming factors, the samples were passed through 10-KDa cutoff Centricons. The intensity of the color was monitored over time, and then, the reaction was stopped and measured at 470 nm.

## Graph plotting and statistical analysis

All graphs were plotted using Prism (version-7). Statistical analysis was performed using the same software. Statistical difference between two groups was determined by unpaired two-tailed Student's *t*-test or one-way ANOVA, as stated in the respective figures. *P*-value < 0.05 was considered significant, and different levels of significance were expressed as follows: *$P < 0.05$; **$P < 0.01$; ***$P < 0.001$; ****$P < 0.0001$.

Expanded View for this article is available online.

## Acknowledgements

This study was supported by the Deutsche Forschungsgemeinschaft (SFB 1118 & GRK 1874-DIAMICOM), DZD, and the Helmholtz Cross Program Topic Metabolic Dysfunction and the Foundation for Diabetes Research. We also thank all members of our group for their support, Dr. Rainer Peperkok, Dr. Stefan Terjung, Dr. Sabine Reither, and other members of ALMF-EMBL Heidelberg for their support and suggestions.

## Author contributions

VK and PPN conceptually designed the experiments; VK, SK, RA, AP performed experimental work. MH and SKo performed clinical studies and analyzed the associated data. ORB, MAM, and SH provided useful resources as well as important inputs for functional studies. VG and AS provided the NHEJ-reporter cell line as well as useful inputs in preparing the manuscript. VK and PPN wrote the manuscript. All authors read this manuscript and agreed for the final submission.

## Conflict of interest

The authors declare that they have no conflict of interest. Correspondence and requests for materials should be addressed to P.P.N or V.K (peter.-nawroth@med.uni-heidelberg.de) or (varun.kumar@med.uni-heidelberg.de).

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
