## [Review Process File · The EMBO Journal]

Compromised DNA repair is responsible for diabetes-associated organ fibrosis

Kumar Varun, Agrawal Raman, Pandey Aparamita, Kopf Stefan, Hoeffgen Manuel, Kaymak Serap, Bandapalli Obul Reddy, Gorbunova Vera, Seluanov Andrei, Mall Marcus A, Herzig Stephan, Nawroth Peter P

Review timeline:

Submission date:	17th Sep 2019
Editorial Decision:	28th Oct 2019
Revision received:	25th Jan 2020
Editorial Decision:	17th Feb 2020
Revision received:	27th Feb 2020
Accepted:	8th Mar 2020

Editor: Daniel Klimmeck

Transaction Report:

1st Editorial Decision

28th Oct 2019

Thank you for the submission of your manuscript (EMBOJ-2019-103477) to The EMBO Journal. Please accept my sincere apologies for the delay with the peer-review of your manuscript. Your manuscript has been sent to two reviewers, please find their reports enclosed below.

Given the referees' recommendations, I would like to invite you to submit a revised version of the manuscript, addressing their concerns. I should add that it is EMBO Journal policy to allow only a single round of revision, and acceptance of your manuscript will therefore depend on the completeness of your responses in this revised version.

Referee #1:

This manuscript links reducing sugars to a DNA repair dysfunction in the context of diabetes-induced lung and kidney fibrosis and pathology. The authors propose a novel potential therapeutic strategy aiming to promote DNA repair in affected tissues by taking advantage of an AAV strategy to transduce a phospho-mimicking nRAGE. The authors show here several interesting findings; 1) Extended exposure to high concentration of reducing sugars induces accumulation of non-repaired DNA damage; 2) STZ-induced diabetes triggers increased DNA damage, senescence and IL6 production in the lung and kidney; 3) STZ-induced lung and kidney DNA damage can be reduced by expressing a mutant nRAGE that mimics ATM phosphorylation. This study is in general technically sound and well conducted. This reviewer has, however, some major critics and concerns that need to be addressed to improve the quality of this manuscript.

1. Overall, the text of the manuscript is somehow confusing and would benefit from extensive editing. In addition, the results section is written in a way that makes the reader have difficulty determining what is actually a novel finding and what has been described in the literature. Moreover, general statements in the manuscript need to be moderated to avoid over-interpretation of the data. For instance, the authors often refer to "diabetic complications" whereas the only things that are experimentally measured relate to lung and kidney DNA damage, senescence and fibrosis. T1D and T2D are very complex disease with a wide range of complications that are most certainly not limited to the kidney and lung dysfunction. As it is, the text is misleading and should be modified to better reflect the findings of the authors, as retinopathy, neuropathy and cardiovascular diseases are also diabetic complications. Finally, the authors state that their findings relate to T1D and T2D. However the mouse model used is a T1D model. The authors would need a T2D model to validate that their findings are directly linked to elevated circulating sugar levels.
2. From data in Fig. 4 and EV9A-C the authors conclude that diabetes associated persistent DNA-DSB signaling affects both lungs and kidneys equally. Lung function and fibrosis were assessed, but kidney function was not measured (serum creatinine for example). Only fibrosis was similar in the two organs. If the albuminuria described in Fig. EV10 is used as a measure of kidney function, then the authors should consider that this parameter was not rescued with their DNA damage reducing approach (nRAGE transduction).
3. The authors use extremely high concentrations of reducing sugars. Indeed, a 30 mM glucose concentration would be abnormally high, even in a diabetic patient (post-prandial glucose 12-20). Circulating fructose levels are more than 20 times lower than circulating glucose levels (PMID: 25100436). Ribose levels are of about 30 μ M in normal and STZ treated rats (PMID: 29033370). The authors should reproduce the experiments using more "relevant" concentrations of reducing sugars (moreover, the authors state that some of their high sugar concentrations induce cell death in 5 days). Images from fructose treated cells (corresponding to figs EV4E and 2B) are not shown whereas all other conditions are shown.
4. To formally link the observed DNA repair abnormalities to the NAD⁺/NADH pool the authors could perform the experiments using antioxidants, like NAC.
5. To demonstrate that reducing sugars trigger senescence and SASP in a cell autonomous manner due to increased DNA damage (and whether this phenomenon may be reduced by the use of antioxidants like NAC) the authors need to measure senescence-associated beta-gal and (at least) IL6 production.
6. The authors should provide the blood glucose level follow-up for the mice used in this project.

Minor comments:

- Fig4D and 4E are cited as 1D and 1E in the results section.
- The X axis in Figure EV10 has no title.
- Fig EV8 is not properly referred to in the results section.
- The resolution of fig 4A needs to be increased.
- The comparisons used for the statistics are sometimes unclear (example fig EV11D).
- Some panels lack statistics (example fig 3B).
- In the "primary fibroblast culture" section, the authors state that they use "respectively genotyped mice," however, only WT cells are used in this manuscript.
- The part of the discussion describing the role of nRAGE in the innate immune system and inflammation, and how it relates to the findings described in this manuscript should be rewritten as it is confusing in its current form. For instance, the authors say " Furthermore, the RAGE transduction was initiated after the progression of fibrosis, thus treatment does not prevent, but rather reverse fibrosis", however they have not tested if their treatment can actually prevent the onset of fibrosis. Moreover, when looking at their results, the fibrotic phenotype appears reduced, not reversed.

Referee #2:

The authors studied potential molecular mechanisms underpinning diabetic complications. They show that DNA repair is impaired in type 1 and type 2 diabetes, which leads to senescence, inflammation, and fibrosis in several organs. They also show that nuclear overexpression of phosphomimetic RAGE in an animal model of diabetes restores DNA repair resulting in reduced DNA damage, inflammation, and fibrosis and restoration of organ function.

The authors first show that normally in mice, an intense mitochondriogenesis takes place in lungs during the first postnatal weeks along with increased ROS production, which is paralleled by DNA damage response (DDR), decreased cell proliferation, production of the proinflammatory cytokine IL-6, increase in the NAD⁺/NADH ratio, and activation of pATM-mediated DNA repair. By the end of the second postnatal week some of these changes (i.e., ATM activation and reduced proliferation) are over whereas other changes (i.e., signs of DNA damage, high ROS and IL-6 levels, and high NAD⁺/NADH ratio) are not. The authors put these findings in relation to the exposure of pups and their subsequent adaptation to atmospheric O₂. Based on these results, the authors moved to a cell model (A549 cells) and an animal model of diabetes (STZ-induced diabetes) to test the hypothesis that stimulation of DNA repair in cells exposed to high glucose might reduce/reverse the detrimental effects of hyperglycemia. The authors show that prolonged exposure of cultured cells to high glucose (and fructose and ribose as well) resulted in DNA damage, reduced NAD⁺/NADH ratio, defective DNA repair, and increase in the fraction of DBC1 bound to PARP, effects that were significantly reversed by etoposide or camptothecin which are drugs able to promote DNA repair. Similar results were obtained in diabetic mice, in which hyperglycemia also caused lung and kidney fibrosis and significant changes in respiratory parameters. Lastly, based on their previous work (Ref. Kumar, Fleming et al. 2017), the authors transduced in diabetic mice a phosphomimetic RAGE mutant that targets the nucleus: they observed that phosphomimetic RAGE but not a non-phosphorylatable RAGE mutant, stimulated DNA repair thus improving the lung and kidney histology (reduced fibrosis) and respiratory parameters, though not albuminuria. The authors conclude that "metabolite induced DNA damage, DDR and persistent DNA damage signaling are the common soil for several complications of diabetes" and that novel therapies, including phosphomimetic RAGE might be envisaged "not to prevent, but rather to reverse diabetes induced organ fibrosis and dysfunction".

The most interesting finding in this work is that transduction of diabetic mice with a phosphomimetic RAGE mutant reverses diabetes-induced organ fibrosis and dysfunction via stimulation of repair of ROS-induced DNA damage. The importance of this results is twofold: because of its potential clinical application and because a "good" side of RAGE in diabetes complications has been revealed. So far, RAGE has been viewed as an important molecular determinant of diabetes complications, transducing effects of a number of ligands as a transmembrane (pattern recognition) receptor. In their previous work (Ref. Kumar, Fleming et al. 2017), the authors showed that i) loss of RAGE (Ager^{-/-} mice) was causally linked to perpetual double-strand breaks (DSB) signaling, cellular senescence and fibrosis; ii) reconstitution of RAGE efficiently restored DSB-repair and reversed pathological anomalies in a mouse model of idiopathic pulmonary fibrosis (Ager^{-/-} mice); and iii) this RAGE's activity was contingent on ATM-dependent RAGE phosphorylation and nuclear localization of phosphorylated RAGE. Whereas a role for phosphorylated RAGE in promoting DNA repair has been documented (ibid.), the finding that phosphomimetic RAGE reverses tissue fibrosis in an animal model of diabetes is novel. The conclusions of the paper are justified based on the presented data, however a few points need to be clarified (see below).

Comments

Minor points

1. Abstract. The word "men" should be replaced by the word "humans".
2. Fig. Ev1B. Should P15 be P14?
3. Fig. 3B. p values are not indicated.
4. Pages 11 and 12. Fig. 1D and Fig. 1E should be Fig. 4D and Fig. 4E, respectively.
5. Page 12. "analysis of lungs from STZ mice also showed significant depletion of NAD⁺/NADH pool (Fig 4F)". As it stands, this sentence is misleading. I suggest to rephrase as follows "analysis of lungs from STZ mice also showed a significant decrease in the NAD⁺/NADH ratio (Fig 4F)".
6. Subheading "Restoring DNA-repair potential ameliorates senescence; senescence associated secretory phenotype and reverses fibrosis". Please replace ";" with "and".

7. Fig. 5A. The 2-5 numbering should be changed to 1-5.
 8. Discussion, page 14, third last line. I suggest to replace the word "change" with the word "decrease".

Major points

1. How were negative controls in Figs. 1E, 5A, EV2B,C, EV11C and EV12B obtained?
2. Page 9. "increased concentrations of reducing sugars shifted the equilibrium towards a decreased NAD⁺ and increasing NADH cofactor pool". That high reduced sugar concentrations decrease the NAD⁺/NADH ratio is not a novel observation (J Diabetes Res 2014:137919, 2014). In hyperglycemic conditions, the amount of NADH produced exceeds the electron transport chain capacity leading to an increase in the NADH/NAD⁺ ratio that is known as reductive stress. Indeed, overnutrition increases H₂O₂ generation by mitochondria to relieve the reducing pressure created by fuel overload (Trends Endocrinol Metab 23: 142-153, 2012); an excess of NADH imposes a stress on the electron transport chain resulting in electron leakage that causes an improper O₂ reduction to superoxide anion. The cell responds by activating the transcription of antioxidants via Nrf2 activity, which increases cellular levels of GSH further contributing to reductive stress. Moreover, in hyperglycemic conditions excess glucose fluxes into the polyol pathway in which glucose is reduced by the NADPH-dependent enzyme, aldose reductase, to form sorbitol that is then converted to fructose by NAD⁺-dependent sorbitol dehydrogenase (Diabetes 54: 1615-1625, 2005). In a highly reduced environment (i.e., decreased NAD⁺/NADH ratio) mitochondrial ROS production is so high and fast that it overcomes the cell's ROS scavenging capacity, even though the latter should be maximally potentiated (Biochim Biophys Acta 1797: 865-877, 2010; Biochim Biophys Acta 1837: 287-295, 2014), which ultimately leads to ROS-induced cell damage (Biochim Biophys Acta 1847: 514-525, 2015). This suggests that reductive stress can be detrimental by paradoxically leading to oxidative stress. The authors should take these findings into consideration when describing/discussing the results reported herein.
3. Blots in Fig. EV6A (lung adenocarcinoma [A549] cells) are the same as in Fig. 2F (primary murine lung fibroblasts). A mistake during the preparation of the two Figures?
4. Fig. EV11C,D. Contrary to control lung, nuclei of diabetic lung are intensely positive for the DNA DSB marker pATM as are nuclei of lung of mice injected with non-phosphorylatable RAGE mutant, pointing to defective DNA repair; however injection with phosphomimetic RAGE results in a robust reduction of pATM staining. These results would put pATM downstream of phosphomimetic RAGE. However, in their previous work (Kumar, Fleming et al. 2017) the authors show that "RAGE phosphorylation by ATM is central for repair of radiation induced DNA damage and of tissue fibrosis and cancer formation" and "ATM-phosphorylated nuclear RAGE prevents cellular senescence, inflammation, pulmonary fibrosis and cancer". Thus, while ATM is required for RAGE phosphorylation and nuclear localization, transduction with phosphomimetic RAGE robustly reduces ATM activity. Can one envisage a scenario in which ATM phosphorylates RAGE and phosphorylated RAGE promote DNA repair, which leads to ATM inactivation?
5. Discussion, page 15 "it remains unknown, how ROS are generated, since an increase in ROS might be caused by changes in carbohydrate metabolism (Lorenzi, Montisano et al. 1986)". Please see point 1 above.
6. Reduction of tissue fibrosis upon transduction with phosphomimetic RAGE suggests that phosphomimetic RAGE strongly interferes with the molecular machinery that causes fibrosis. Is this a result of improved DNA repair in lung- and kidney-specific cell types exclusively or is transduced phosphomimetic RAGE also acting in macrophages (i.e. the cell type principally responsible for the liberation of fibrosis-inducing factors) and/or fibroblasts (i.e. the cell type principally responsive to fibrosis-inducing factors)? In other words, is phosphomimetic RAGE activity in lung- and kidney-specific cell types enough to improve tissue histology and function (albeit via a mechanism to be identified)? While the authors show that transduction with phosphomimetic RAGE reduces mean fluorescence intensity of the proinflammatory IL-6 in lung tissue (Fig. 5D), measurements of serum levels of proinflammatory and antiinflammatory cytokines and/or tissue levels of proinflammatory and antiinflammatory cytokine mRNAs would be advisable to support the authors' conclusion that phosphomimetic RAGE reduces inflammation.

Please see next page.

Referee #1:

This manuscript links reducing sugars to a DNA repair dysfunction in the context of diabetes-induced lung and kidney fibrosis and pathology. The authors propose a novel potential therapeutic strategy aiming to promote DNA repair in affected tissues by taking advantage of an AAV strategy to transduce a phospho-mimicking nRAGE. The authors show here several interesting findings; 1) Extended exposure to high concentration of reducing sugars induces accumulation of non-repaired DNA damage; 2) STZ-induced diabetes triggers increased DNA damage, senescence and IL6 production in the lung and kidney; 3) STZ-induced lung and kidney DNA damage can be reduced by expressing a mutant nRAGE that mimics ATM phosphorylation. This study is in general technically sound and well conducted. This reviewer has, however, some major critics and concerns that need to be addressed to improve the quality of this manuscript.

Question 1(i). Overall, the text of the manuscript is somehow confusing and would benefit from extensive editing. In addition, the results section is written in a way that makes the reader have difficulty determining what is actually a novel finding and what has been described in the literature. Moreover, general statements in the manuscript need to be moderated to avoid over-interpretation of the data. For instance, the authors often refer to "diabetic complications" whereas the only things that are experimentally measured relate to lung and kidney DNA damage, senescence and fibrosis. T1D and T2D are very complex disease with a wide range of complications that are most certainly not limited to the kidney and lung dysfunction. As it is, the text is misleading and should be modified to better reflect the findings of the

authors, as retinopathy, neuropathy and cardiovascular diseases are also diabetic complications.

Response: We thank the reviewer for this comment and suggestion to improve the text which can be misleading for the readers. Thus in order to convey a more specific message, we edited the manuscript title and text from diabetic complications to diabetes-associated organ fibrosis. The title of the edited version of the manuscript now reads to “A maladaptive metabolic re-programming compromises the DNA repair potential of cells and is responsible for diabetes-associated organ fibrosis”. The respective changes in the main manuscript are marked in red.

Question 1(ii). Finally, the authors state that their findings relate to T1D and T2D. However the mouse model used is a T1D model. The authors would need a T2D model to validate that their findings are directly linked to elevated circulating sugar levels.

Response: We thank the reviewer for this comment and suggestion to verify the direct role of elevated sugar levels in DNA repair signaling in the type-2 diabetic model also. Here to answer this we used 4-months *db/db* mice as the type-2 diabetes model. In this model, the glucose levels start rising at around 1.2 to 1.6 months of age as compared to the age-matched lean (+/*db*) controls. At 4-months the weight of obese *db/db* mice was around 42 to 51 gms, whereas the lean controls of the same age weighed around 19 to 24 gms. These mice were maintained under normal chow diet. As studied in STZ diabetic model, we used γ H2AX as a marker of DNA-DSBs. Here it was observed that the levels of γ H2AX were markedly enhanced in both lungs (**Figure 1R Q1(ii) A-B**) and kidney (**Figure 1R Q1(ii) C-D**) of *db/db* mice as compared to the lean controls(+/*db*), shows elevated DSBs signaling. Furthermore to test if these DSBs mark these organs for persistent damage signaling, we studied cellular senescence staining in these organs. Here we observed that senescence, as

marked by β -galactosidase staining was also markedly enhanced in obese *db/db* mice (**Figure 1R Q1(ii) E**) as compared to the lean controls (*+/db*). Moreover persistent DNA damage signaling is known to affect the morphological as well as physiological integrity of the organs (Armanios, Chen et al. 2007, Rodier, Coppe et al. 2009). To test if this is the case with *db/db* model too, we used classical H&E and Masson's Trichrome staining on lung and kidney collected as described above. Our result shows that both *db/db* lung and kidney shows marked disruption in morphology as well as accumulation of ECM components (**Figure 1R Q1(ii) F-G**; deposited ECM in blue) as compared to the age-matched lean controls (*+/db*). These results clearly showed that elevated blood glucose levels compromise the DNA repair potential of these organs and the persistent DNA-DSBs signaling coupled to senescence is the main culprit for organ fibrosis in both type-1 and type-2 diabetic models presented here. This new data is now included in the main manuscript under Fig EV 11 and Fig EV13 and the respective methods were also modified accordingly.

Figure 1R Q1(ii)

Figure 1R Q1(ii) continued

Figure 1R Q1 (ii): (A) Representative immunoblots of lungs harvested from 4-months old *+/db* (non-diabetic lean control) or *db/db* mice and probed for γ H2AX. Histone-H3 was used as a loading control. (B) Representative images of lungs from 4-months old non-diabetic lean controls (*+/db*) and age-matched obese *db/db* mice, showing DNA damage

foci, as marked by γ H2AX (in red). Here tubulin was used as morphology marker (shown in green) and blue nuclear staining represents DAPI (Scale 10 μ m). The zoomed portions of each image were shown in thick white lines in the upper left corner. (C) Representative immunoblots of kidneys harvested from old 4-months *+/db* (non-diabetic control) or *db/db* mice and probed for γ H2AX. Histone-H3 was used as a loading control. (D) Representative images of kidneys from 4-months old non-diabetic lean controls (*+/db*) and age-matched obese *db/db* mice, showing DNA damage foci, as marked by γ H2AX (in red). Here tubulin was used as morphology marker (shown in green) and blue nuclear staining represents DAPI (Scale 10 μ m). The zoomed portions of each image were shown in thick white lines in the upper left corner of each image. (E) Representative images of cellular senescence staining in lung and kidney of 4-months old lean controls (*+/db*) versus obese *db/db* diabetic mice. Sections stained for cellular senescence-associated β -galactosidase [β -Gal] as described in Methods and visualized by bright field and polarized light, The senescent areas are recognized by its bluish staining (Scale 40 μ m). Eosin (pinkish-red) was used as Morphology stain. (F) Representative images of H&E stained lung or kidney sections from 4-months old lean controls (*+/db*), or age-matched *db/db* diabetic mice, as described in Methods. H&E staining was visualized by bright field and polarized light (Scale 40 μ m). (G) Representative images of lungs (upper panel) and kidneys (lower panel) from 4-months old lean controls (*+/db*), or age-matched *db/db* diabetic mice stained with Masson's Trichrome stain, as described in Methods and visualized by bright field and polarized light, ECM is recognized by its blue staining (Scale 40 μ m).

Question 2. From data in Fig. 4 and EV9A-C the authors conclude that diabetes associated persistent DNA-DSB signaling affects both lungs and kidneys equally. Lung function and fibrosis were assessed, but kidney function was not measured (serum creatinine for example). Only fibrosis was similar in the two organs. If the albuminuria described in Fig. EV10 is used as a measure of kidney function, then the authors should consider that this parameter was not rescued with their DNA damage reducing approach (nRAGE transduction).

Response: We thank the reviewer for this comment and for providing us the opportunity to clarify this. Here we completely agree that albuminuria is not the right parameter to assess the kidney fibrosis. The disconnect between albuminuria, a surrogate of vascular damage, and fibrosis observed in our remission cohort, but also reported by others too (Magalhaes, Pejchinovski et al. 2017). The patient data

presented in figure EV10 (Revised figure EV14) was used to simulate the link between diabetic lung and renal fibrosis and because of stringent clinical ethical norms, we only have albuminuria to relate to renal function and fibrosis. As the functional tests for lung function (such as spirometry and plethysmography) are straight forward, hence it is easier to perform them in patients, than doing a potentially dangerous renal biopsy. Similarly, very high-resolution CT/MRI scans (Friedli, Crowe et al. 2016), which not only require one more specific visit to the ward but, also can have its own implications as well, thus most of the patients deny taking this extra test. Therefore we are limited to albuminuria as a surrogate for vascular renal damage. But in order to evaluate the functional changes associated with transduced kidney in mice, we present two functional tests which show that the kidney function has been improved in the phosphomimetic RAGE mutant (RAGE-EE) cohort compared to the non-phosphorylatable RAGE (**Figure 1R Q2**). Further, as we are aware of previous reports (Magalhaes, Pejchinovski et al. 2017, Rauchman and Griggs 2019), thus we don't state that these tests are specific to fibrosis, but rather help to evaluate the renal function of the transduced mice. Our data are therefore also complemented with other data showing reduced DNA damage signaling, SASPs, senescent and fibrotic areas.

The first functional assay, as suggested by you, measured the levels of excreted creatinine in urine from control or transduced groups. Here we observed a significant change in the excreted creatinine in the group transduced with phospho-mimetic RAGE mutant (RAGE-EE), but not in the group transduced with non-phosphorylatable RAGE mutant (RAGE-AA) (**Figure 1R Q2 A**). We include this additional creatinine data to study the kidney function in our transduced study cohort. The 2nd parameter we used was the changes in the total volume of urine excretion. Since diabetic mice excrete a large volume of urine and we plotted the total volume of urine excreted in our transduction cohort within 24 hours. We also observed a

significant reduction in the total volume of urine excreted in 24 hours in the mice transduced with phospho-mimetic RAGE mutant (RAGE-EE) compared to the non-phosphorylatable RAGE mutant (RAGE-AA) (**Figure 1R Q2 B**).

Both these functional results support our other results, showing that RAGE-EE treatment timely “turns-off” the persistent DNA damage signaling, thus reducing the organ fibrosis and improving the physiology of the organ studied.

These new data was incorporated into the revised manuscript under figure EV 16E-F.

Figure 1R Q2: (A) Quantitative analysis of the urinary creatinine in urine from 6-month STZ induced diabetic mice, transduced with AAV2/8 as described in Figure EV14A (mean \pm SD, *: $p < 0.05$, **: $p < 0.01$, $N = 8$). Data presented in this Figure, as well as in Figure 5 and EV Figure 14 is from the same mice. (B) Quantitative analysis of the total urinary volume excreted in 24 hours from 6-month STZ induced diabetic mice, transduced with AAV2/8 as described in Figure EV14A (mean \pm SD, **: $p < 0.01$, $N = 8$). Data presented in this Figure, as well as in Figure 5 and EV Figure 14 is from the same mice.

Question 3. (i) The authors use extremely high concentrations of reducing sugars. Indeed, a 30 mM glucose concentration would be abnormally high, even in a diabetic patient (post-prandial glucose 12-20). Circulating fructose

levels are more than 20 times lower than circulating glucose levels (PMID: 25100436). Ribose levels are of about 30 μ M in normal and STZ treated rats (PMID: 29033370). The authors should reproduce the experiments using more "relevant" concentrations of reducing sugars (moreover, the authors state that some of their high sugar concentrations induce cell death in 5 days).

Response: We thank the reviewer for this comment and for providing us the opportunity to clarify this. Previously we used a high concentration of glucose as well as other reducing sugars, just to mimic the uncontrolled diabetic condition within a short duration (like 3-5 days) and we agree that these concentrations were not ideal. Thus (as suggested by you) we used a mean prandial concentration of glucose, which is also similar to the blood glucose levels of our STZ cohort (high glucose: 17mM)(Sidhu, Nundy et al. 2001, Wang, Smyl et al. 2018) with or without the combination of other reducing sugars (Fructose: 1mM; Ribose: 100 μ M)(Laughlin 2014, Chen, Su et al. 2017). The low glucose concentration was kept the same (5.5mM). In the first experiment, we studied the etoposide (5 μ M for 60 minutes) induced DNA repair kinetics (for 2, 8 or 24 hours after damage) in A549 cells stimulated with glucose for 5, 10 or 15 days. Here it was observed that high glucose (17mM) condition affects the DNA repair potential, as marked by the DNA damage marker γ H2AX. The low glucose cultivated cells can repair its DNA-DSBs within 24 hours, whereas the high glucose cultivated ones failed to do so (**Figure 1R Q3(i) A**). Here Histone-H3 was used as a loading control. Furthermore, the ability of cells to repair the etoposide-induced DSBs was inversely related to the duration of pre-stimulation (5, 10 or 15 days) as compared to the controls (low glucose concentration; 5.5mM). Furthermore, the repair difference was very much evident at the 5-day stimulation stage too, allowed us to use this stimulation duration for studying the effects of other reducing sugars (fructose and/or ribose along with indicated glucose concentrations). The DSBs were induced by Etoposide as described above and the

repair kinetics was studied by using γ H2AX. The stimulation of fructose and/or ribose further decreases the DNA repair capacity of the cells maintained under only high glucose (**Figure 1R Q3(i) B**). In order to mark this observation as a general rather than a cell-specific event, we verified it using kidney-specific cells too (**Figure 1R Q3(i) C, D**). Thus our cell-based study conclusively points towards a decreased DNA repair potential of cells maintained under hyperglycemic conditions.

These new data was incorporated into the revised manuscript under Fig EV 6A-D.

Figure 1R Q3(i)

Figure 1R Q3(i): (A) Representative immunoblot from lysates of A549 cells cultured in the presence of the indicated glucose concentrations for 5, 10 or 15 days and treated with etoposide ($5\mu\text{M}$ for 60 minutes) were probed for the DNA-DSBs marker γ H2AX. Histone-H3 was used as a loading control. (B) Representative immunoblot from lysates of A549 cells cultured in the presence of the indicated glucose concentrations along with or without fructose and/or ribose for 5 days and treated with etoposide ($5\mu\text{M}$ for 60 minutes) were probed for the DNA-DSBs marker γ H2AX after the indicated intervals. Histone-H3 was used as a loading control. (C) Representative immunoblot from lysates of HEK-293

cells cultured in the presence of the indicated glucose concentrations for 5, 10 or 15 days and treated with etoposide (5 μ M; upper panel) were probed for the DNA-DSBs marker γ H2AX. Histone-H3 was used as a loading control. **(D)** Representative immunoblots from lysates of HEK-293 cells cultured in the presence of the indicated glucose concentration along with or without fructose and/or ribose for 5 days and treated with etoposide (5 μ M; upper panel) were probed for the DNA-DSBs marker γ H2AX after the indicated intervals. Histone-H3 was used as a loading control.

Question 3.(ii) Images from fructose treated cells (corresponding to figs EV4E and 2B) are not shown whereas all other conditions are shown.

Response: We thank the reviewer for this comment and for the suggestion to present not just the quantification, but also the image data from fructose treated cells. Earlier, because of space limitations (particularly in the main figure 2), we omit the image data from fructose stimulated cells. Now we carefully reformatted the figures and incorporated this image data and the composite movie M1.

Below we are presenting the image data in the reviewer's file under **Figure 1R Q3 (ii) A, B, and C** (for Laser, etoposide or camptothecin induced DNA damage respectively).

In the revised version of the manuscript these data are presented in Figure 2C, EV3 (B) and EV4 (E) [for laser, etoposide or camptothecin induced DNA damage respectively]. In addition, the revised movie M1 has now data from fructose treated cells too (showing live recruitment kinetics of hPARP-mCherry and hXRCC4-GFP).

Figure 1R Q3 (ii)

Figure 1R Q3(ii): (A) Still images of A549 cells showing the live recruitment of hPARP-mCherry and hXRCC4-GFP, at the site of laser-induced DNA-DSBs from the cells pre-treated with the indicated reducing sugars for 3 days as described in methods. (B) Immunofluorescence analysis of DSB associated foci, marked by γ H2AX in lung adenocarcinoma (A549) cells, cultured in fructose (40mM) for 3 days and treated with etoposide ($5\mu\text{M}$ for 60 minutes). The resolution of DNA-DSB foci, as marked by γ H2AX was monitored over 24 hours after drug treatment (Scale $10\mu\text{m}$). (C) Immunofluorescence analysis of DSB associated foci, marked by γ H2AX in lung adenocarcinoma (A549) cells, cultured in fructose (40mM) for 3 days and treated with camptothecin ($1\mu\text{M}$ for 60 minutes). The resolution of DNA-DSB foci,

as marked by γ H2AX was monitored over 24 hours after drug treatment (Scale 10 μ m). CML (marked in green) served as induction control.

Question 4. To formally link the observed DNA repair abnormalities to the NAD⁺/NADH pool the authors could perform the experiments using antioxidants, like NAC.

Response: We thank the reviewer for this comment and suggestion. As in cell culture, the nicotinamide adenine di-nucleotide equilibrium can be modified by supplementing the growth medium accordingly. To answer this question we used the previously described method which can be used to alter the NAD⁺/NADH equilibrium (Ma, Chen et al. 2011). Here we pre-treated the A549 cells growing under normal glucose condition (5.5mM), with either 100 μ M or 250 μ M NADH for 24 hours and the DNA-DSBs repair (Etoposide 5 μ M for 60 minutes) kinetics was studied. Our result shows that shifting the NAD⁺/NADH equilibrium towards NADH side, affects the DNA repair kinetics as marked by DNA-DSBs repair marker γ H2AX (Figure 1R Q4 A). Hence the cells treated with NADH simulate the conditions similar to the prolonged hyperglycemia environment even though these cells were maintained under normal glucose conditions (5.5mM) validates the importance of high glucose in perturbing this redox balance. We further validated this observation in HEK-293 also shows that this NADH mediated effects are not limited to only lungs (Figure 1R Q4 B). Hence this further justifies our hypothesis that high glucose concentration affects the DNA repair potential by disturbing the NAD⁺/NADH equilibrium. This additional data, as well as reference, was incorporated into the revised version of this manuscript under figure (EV7 C and D).

The N-Acetyl-cysteine (NAC) part is answered in next question.

Figure 1R Q4

Figure 1R Q4: (A) Lung adenocarcinoma (A549) cells cultured in low glucose (5.5mM) medium containing no or additional NADH (100µM or 250µM) for 24 hours and then treated with etoposide (5µM for 60 minutes). The resolution of DNA-DSBs (marked by γ H2AX) was monitored over indicated time points. Histone-H3 was used as a loading control. (B) HEK-293 cells cultured in low glucose (5.5mM) medium containing no or additional NADH (100µM or 250µM) for 24 hours and then treated with etoposide (5µM for 60 minutes). The resolution of DNA-DSBs (marked by γ H2AX) was monitored over indicated time points. Histone-H3 was used as a loading control.

Question 5. To demonstrate that reducing sugars trigger senescence and SASP in a cell autonomous manner due to increased DNA damage (and whether this phenomenon may be reduced by the use of antioxidants like NAC) the authors need to measure senescence-associated beta-gal and (at least) IL6 production.

Response: We thank the reviewer for this comment and suggestion. To prove that the cellular senescence and SASP, associated with STZ model is linked to persistent DNA damage signaling, we followed the experiments from Question 3(i) [by using the mean prandial concentration of glucose (17mM), fructose (1mM) and ribose (100µM) (Laughlin 2014, Chen, Su et al. 2017). Cells maintained under hyperglycemic conditions show markedly unrepaired DNA DSBs and thus it is important to distinguish if this unrepaired DNA damage could impose an essentially irreversible growth arrest and if the antioxidant therapies can be beneficial under these situations. To test this A549 cells were pretreated with indicated sugars for 5 days (with or without N-Acetyl cysteine; NAC; 2mM (You, Shin et al. 2014)) and the DNA damage

was induced by treating them with Etoposide (5 μ M for 60 minutes). After the damage, cells were allowed to repair its DNA-DSBs and senescence and SASPs development was monitored in these cells after 10 days of damage treatment as suggested previously (Rodier, Coppe et al. 2009). It was observed that cells maintained under hyperglycemic conditions showed elevated cellular senescence, as evidenced by β -galactosidase activity. Furthermore, the addition of fructose and ribose under these hyperglycemic conditions further enhanced the levels of it (**Figure 1R Q5 A**). The cells maintained under low glucose conditions show very little senescence, which can be slightly induced by the presence of fructose and ribose. Remarkably NAC treatment completely abolishes the cellular senescence in cells maintained under low glucose conditions, but only slightly reduced it in the cells maintained under hyperglycemic conditions (**Figure 1R Q5 A**). This observation was further verified by quantitative analysis of SASP marker IL-6 from the culture supernatant collected from the treatments described above (**Figure 1R Q5 B**) also supports the hypothesis that elevated levels of reducing sugars are causatively linked to DSBs associated persistent DNA damage signaling. Concurrently this hyperglycemia linked, senescence and SASP was also confirmed in kidney-specific cells also (**Figure 1R Q5 C, D**), further supports the notion that unrepaired DNA-DSBs in hyperglycemia cumulatively activates the persistent DNA damage signaling, which then modulates the cell cycle, senescence and SASP. In the revised version of this manuscript, this data was included under Fig EV8 A-D.

Figure 1R Q5

(A)

(B)

(C)

(D)

Figure 1R Q5: (A) Representative images of A549 cells cultured in the presence of the indicated glucose concentrations for 5 days alone or with other reducing sugars (here F indicates for fructose at concentration 1mM, and R indicates for Ribose at concentration 100 μ M) along with NAC (N-Acetyl cysteine at 2mM) and treated with etoposide (5 μ M for 60 minutes), washed and cultivated in the sugars as indicated for 10days and stained for β -galactosidase as indicated in methods. Eosin was used as morphology stain (Scale 40 μ m). (B) Quantitative analysis of the IL-6 secreted in the culture supernatant of A549 cells treated as described in figure EV8A. (mean \pm SD, *: p<0.05). (C) Representative images of murine podocytes cultured in the presence of the indicated glucose concentrations for 5 days alone or with other reducing sugars (here F indicates for fructose at concentration 1mM, and R indicates for Ribose at concentration, 100 μ M) along with NAC (N-Acetyl cysteine at 2mM) and treated with etoposide (5 μ M for 60 minutes), washed and cultivated in the sugars as indicated for 10days and stained for β -galactosidase as indicated in methods. Eosin was used as morphology stain (Scale 40 μ m). (D) Quantitative analysis of the IL-6 secreted in the culture supernatant of murine podocytes treated as described in figure EV8C. (mean \pm SD, *: p<0.05, **: p<0.01).

Question 6. The authors should provide the blood glucose level follow-up for the mice used in this project.

Response: We thank the reviewer for this comment. In order to prevent excessive and potentially lethal hyperglycemia, the diabetic mice which show glucose levels above 500mg/dl (27mM), received 1-2 units of Insulin and upon administration of this dose, the glucose levels can be 350 to 450 mg/dl (19 to 25mM) and thereafter it rise accordingly. So our study cohorts were maintained between 350-450mg/dl. We updated this information in the methods section of the revised manuscript.

Minor comments:

- **Fig4D and 4E are cited as 1D and 1E in the results section.**

We thank the reviewer for pointing out this error and also apologize for this mistake. We corrected them in the revised manuscript.

- **The X axis in Figure EV10 has no title.**

We thank the reviewer for pointing out this error and also apologize for this mistake. We corrected this missing axis title in the revised manuscript. In the updated

manuscript the above-mentioned figure number (EV10) has been changed to EV14.

- **Fig EV8 is not properly referred to in the results section.**

We thank the reviewer for this suggestion. We explained this figure more clearly. In the updated manuscript the above-mentioned figure number (EV8) has been changed to EV10.

- **The resolution of fig 4A needs to be increased.**

We thank the reviewer for pointing out this and also apologize for this. We corrected the resolution of this graph in the revised manuscript.

- **The comparisons used for the statistics are sometimes unclear (example fig EV11D).**

We thank the reviewer for pointing out this error and also apologize for this mistake. We corrected them in the revised manuscript. In some graphs where bars are very narrow and thus comparative statistics line marking won't be ideal to show, we mentioned the comparative groups in the figure legend.

- **Some panels lack statistics (example fig 3B).**

We thank the reviewer for pointing out this error and also apologize for this mistake. We corrected this in the revised manuscript.

- **In the "primary fibroblast culture" section, the authors state that they use "respectively genotyped mice," however, only WT cells are used in this manuscript.**

We thank the reviewer for pointing out this error and also apologize for this mistake. We corrected them in the revised manuscript.

• The part of the discussion describing the role of nRAGE in the innate immune system and inflammation, and how it relates to the findings described in this manuscript should be rewritten as it is confusing in its current form. For instance, the authors say " Furthermore, the RAGE transduction was initiated after the progression of fibrosis, thus treatment does not prevent, but rather reverse fibrosis", however they have not tested if their treatment can actually prevent the onset of fibrosis. Moreover, when looking at their results, the fibrotic phenotype appears reduced, not reversed.

We thank the reviewer for this suggestion and for providing us the opportunity to clarify this. We made this statement because, in our previous nuclear RAGE manuscript, we presented the RAGE^{-/-} model shows that in the absence of RAGE (Kumar, Fleming et al. 2017), these mice develop fibrosis and constitutive expression of RAGE helps in reversing it. A similar observation was also highlighted by others too (B, Lawson et al. 2013).

Here in our current manuscript on diabetes, we did not study if RAGE prevents the development of fibrosis, thus to avoid any confusion, we edited the original text to "the RAGE transduction was initiated after progression of fibrosis, thus treatment reduces fibrosis".

References for Reviewer-1

Armanios, M. Y., J. J. Chen, J. D. Cogan, J. K. Alder, R. G. Ingersoll, C. Markin, W. E. Lawson, M. Xie, I. Vulto, J. A. Phillips, 3rd, P. M. Lansdorp, C. W. Greider and J. E. Loyd (2007). "Telomerase mutations in families with idiopathic pulmonary fibrosis." *N Engl J Med* **356**(13): 1317-1326.

B, B. M., W. E. Lawson, T. D. Oury, T. H. Sisson, K. Raghavendran and C. M. Hogaboam (2013). "Animal models of fibrotic lung disease." *Am J Respir Cell Mol Biol* **49**(2): 167-179.

Chen, X., T. Su, Y. Chen, Y. He, Y. Liu, Y. Xu, Y. Wei, J. Li and R. He (2017). "d-Ribose as a Contributor to Glycated Haemoglobin." *EBioMedicine* **25**: 143-153.

Friedli, I., L. A. Crowe, L. Berchtold, S. Moll, K. Hadaya, T. de Perrot, C. Vesin, P. Y. Martin, S. de Seigneux and J. P. Vallee (2016). "New Magnetic Resonance Imaging Index for Renal Fibrosis Assessment: A Comparison between Diffusion-Weighted Imaging and T1 Mapping with Histological Validation." *Sci Rep* **6**: 30088.

Kumar, V., T. Fleming, S. Terjung, C. Gorzelanny, C. Gebhardt, R. Agrawal, M. A. Mall, J. Ranzinger, M. Zeier, T. Madhusudhan, S. Ranjan, B. Isermann, A. Liesz, D. Deshpande, H. U. Haring, S. K. Biswas, P. R. Reynolds, H. P. Hammes, R. Peperkok, P. Angel, S. Herzig and P. P. Nawroth (2017). "Homeostatic nuclear RAGE-ATM interaction is essential for efficient DNA repair." Nucleic Acids Res **45**(18): 10595-10613.

Laughlin, M. R. (2014). "Normal roles for dietary fructose in carbohydrate metabolism." Nutrients **6**(8): 3117-3129.

Ma, Y., H. Chen, W. Xia and W. Ying (2011). "Oxidative stress and PARP activation mediate the NADH-induced decrease in glioma cell survival." Int J Physiol Pathophysiol Pharmacol **3**(1): 21-28.

Magalhaes, P., M. Pejchinovski, K. Markoska, M. Banasik, M. Klinger, D. Svec-Billa, I. Rychlik, M. Rroji, A. Restivo, G. Capasso, F. Bob, A. Schiller, A. Ortiz, M. V. Perez-Gomez, P. Cannata, M. D. Sanchez-Nino, R. Naumovic, V. Brkovic, M. Polenakovic, W. Mullen, A. Vlahou, P. Zurbig, L. Pape, F. Ferrario, C. Denis, G. Spasovski, H. Mischak and J. P. Schanstra (2017). "Association of kidney fibrosis with urinary peptides: a path towards non-invasive liquid biopsies?" Sci Rep **7**(1): 16915.

Rauchman, M. and D. Griggs (2019). "Emerging strategies to disrupt the central TGF-beta axis in kidney fibrosis." Transl Res **209**: 90-104.

Rodier, F., J. P. Coppe, C. K. Patil, W. A. Hoeijmakers, D. P. Munoz, S. R. Raza, A. Freund, E. Campeau, A. R. Davalos and J. Campisi (2009). "Persistent DNA damage signalling triggers senescence-associated inflammatory cytokine secretion." Nat Cell Biol **11**(8): 973-979.

Sidhu, S. S., S. Nundy and R. K. Tandon (2001). "The effect of the modified puestow procedure on diabetes in patients with tropical chronic pancreatitis--a prospective study." Am J Gastroenterol **96**(1): 107-111.

Wang, B., C. Smyl, C. Y. Chen, X. Y. Li, W. Huang, H. M. Zhang, V. J. Pai and J. X. Kang (2018). "Suppression of Postprandial Blood Glucose Fluctuations by a Low-Carbohydrate, High-Protein, and High-Omega-3 Diet via Inhibition of Gluconeogenesis." Int J Mol Sci **19**(7).

You, B. R., H. R. Shin and W. H. Park (2014). "PX-12 inhibits the growth of A549 lung cancer cells via G2/M phase arrest and ROS-dependent apoptosis." Int J Oncol **44**(1): 301-308.

Referee #2:

The authors studied potential molecular mechanisms underpinning diabetic complications. They show that DNA repair is impaired in type 1 and type 2 diabetes, which leads to senescence, inflammation, and fibrosis in several organs. They also show that nuclear overexpression of phosphomimetic RAGE in an animal model of diabetes restores DNA repair resulting in reduced DNA damage, inflammation, and fibrosis and restoration of organ function.

The authors first show that normally in mice, an intense mitochondriogenesis takes place in lungs during the first postnatal weeks along with increased ROS production, which is paralleled by DNA damage response (DDR), decreased cell proliferation, production of the proinflammatory cytokine IL-6, increase in the NAD⁺/NADH ratio, and activation of pATM-mediated DNA repair. By the end of the second postnatal week some of these changes (i.e., ATM activation and reduced proliferation) are over whereas other changes (i.e., signs of DNA damage, high ROS and IL-6 levels, and high NAD⁺/NADH ratio) are not. The authors put these findings in relation to the exposure of pups and their subsequent adaptation to atmospheric O₂. Based on these results, the authors moved to a cell model (A549 cells) and an animal model of diabetes (STZ-induced diabetes) to test the hypothesis that stimulation of DNA repair in cells exposed to high glucose might reduce/reverse the detrimental effects of hyperglycemia. The authors show that prolonged exposure of cultured cells to high glucose (and fructose and ribose as well) resulted in DNA damage, reduced NAD⁺/NADH ratio, defective DNA repair, and increase in the fraction of DBC1 bound to PARP, effects that were significantly reversed by etoposide or camptothecin which are drugs able to promote DNA repair. Similar results were obtained in diabetic mice, in which

hyperglycemia also caused lung and kidney fibrosis and significant changes in respiratory parameters. Lastly, based on their previous work (Ref. Kumar, Fleming et al. 2017), the authors transduced in diabetic mice a phosphomimetic RAGE mutant that targets the nucleus: they observed that phosphomimetic RAGE but not a non-phosphorylatable RAGE mutant, stimulated DNA repair thus improving the lung and kidney histology (reduced fibrosis) and respiratory parameters, though not albuminuria. The authors conclude that "metabolite induced DNA damage, DDR and persistent DNA damage signaling are the common soil for several complications of diabetes" and that novel therapies, including phosphomimetic RAGE might be envisaged "not to prevent, but rather to reverse diabetes induced organ fibrosis and dysfunction".

The most interesting finding in this work is that transduction of diabetic mice with a phosphomimetic RAGE mutant reverses diabetes-induced organ fibrosis and dysfunction via stimulation of repair of ROS-induced DNA damage. The importance of this results is twofold: because of its potential clinical application and because a "good" side of RAGE in diabetes complications has been revealed. So far, RAGE has been viewed as an important molecular determinant of diabetes complications, transducing effects of a number of ligands as a transmembrane (pattern recognition) receptor. In their previous work (Ref. Kumar, Fleming et al. 2017), the authors showed that i) loss of RAGE (Ager^{-/-} mice) was causally linked to perpetual double-strand breaks (DSB) signaling, cellular senescence and fibrosis; ii) reconstitution of RAGE efficiently restored DSB-repair and reversed pathological anomalies in a mouse model of idiopathic pulmonary fibrosis (Ager^{-/-} mice); and iii) this RAGE's activity was contingent on ATM-dependent RAGE phosphorylation and nuclear localization of phosphorylated

RAGE. Whereas a role for phosphorylated RAGE in promoting DNA repair has been documented (ibid.), the finding that phosphomimetic RAGE reverses tissue fibrosis in an animal model of diabetes is novel. The conclusions of the paper are justified based on the presented data, however a few points need to be clarified (see below).

Comments

Minor points

1. Abstract. The word "men" should be replaced by the word "humans".

Response: We thank the reviewer for this suggestion and made the respective change in the revised version of this manuscript.

2. Fig. Ev1B. Should P15 be P14?

Response: We thank the reviewer for pointing out this error and also apologize for this. We corrected this error in the revised version of the manuscript.

3. Fig. 3B. p values are not indicated.

Response: We thank the reviewer for pointing out this error and also apologize for this. We corrected this error in the revised version of the manuscript.

4. Pages 11 and 12. Fig. 1D and Fig. 1E should be Fig. 4D and Fig. 4E, respectively.

Response: We thank the reviewer for pointing out this gross error, which somehow missed from our notice. We also apologize for this mistake. We corrected them in the revised manuscript.

5. Page 12. "analysis of lungs from STZ mice also showed significant depletion of NAD⁺/NADH pool (Fig 4F)". As it stands, this sentence is misleading. I suggest to rephrase as follows "analysis of lungs from STZ mice also showed a significant decrease in the NAD⁺/NADH ratio (Fig 4F). "

Response: We thank the reviewer for this suggestion and made the respective change in the revised version of this manuscript.

6. Subheading "Restoring DNA-repair potential ameliorates senescence; senescence associated secretory phenotype and reverses fibrosis". Please replace ";" with "and".

Response: We thank the reviewer for this suggestion and made the respective change in the revised version of this manuscript. The update subtitle now reads to “Restoring DNA repair potential ameliorates senescence and senescence-associated secretory phenotype as well as reverses fibrosis”

7. Fig. 5A. The 2-5 numbering should be changed to 1-5.

Response: We thank the reviewer for this suggestion and made the respective change in the revised version of this manuscript.

8. Discussion, page 14, third last line. I suggest to replace the word "change" with the word "decrease".

Response: We thank the reviewer for this suggestion and made the respective change in the revised version of this manuscript.

Major points

Question 1. How were negative controls in Figs. 1E, 5A, EV2B,C, EV11C and EV12B obtained?

Response: We thank the reviewer for this comment and for providing us the opportunity to clarify this. In our negative control staining, we did not use the primary antibody or the antibody against the DNA-DSBs marker presented in the panel, but used the antibody dilution/incubation buffer alone during each negative control staining and the rest of steps like incubating with fluorochrome-conjugated secondary antibodies were performed the same way for all the staining.

During the final presentation of the data and to keep the color figure size limited, we showed only one of the images as a mark of presentation as the negative control. Here in figure 1E, we presented postnatal day 14 (P14) as the negative control. But at the same time, we also imaged negative control sections from P1, P3, and P7. Now in (**Figure 2R Q1**), we present the negative control data from other postnatal lungs. The same holds true for figure EV2B and C.

Similarly the negative controls for 5A, EV11C were also obtained from each group simultaneously, but for the purpose of clear presentation we include only control group as the negative control in figure 5A and the transduced group (expressing RFP alone) as the negative control for figure EV11C.

Similarly in the negative control staining for EV12B (revised figure no. EV16B), we did not use the primary antibody (γ H2AX antibody) presented in the panel but used alone antibody dilution/incubation buffer during each staining and rest of the steps like incubating with fluorochrome-conjugated secondary antibodies, washing, DAPI was performed same way.

Further, in order to provide a clear explanation to the future readers, we edited the methods sections for “Tissue sections immunofluorescence” with the followed text: The negative control staining was performed along with the test marker staining, but here we did not use primary antibody or the antibody against the DNA-DSBs marker presented in the panel but used the antibody dilution/incubation buffer alone during each staining. The rest of the steps like incubating with fluorochrome-conjugated secondary antibodies were performed the same way for all the staining. In addition, we also include short descriptive detail of the negative control in each figure legend.

In the revised version of this manuscript, the figure numbers (without any change in data) of EV11C and EV12B has been changed to EV15C and EV16B respectively.

Figure 2R Q1: Representative no primary antibody negative controls immunofluorescence images of post-natal lungs stained for the DNA damage markers shown in Fig 1E and EV 2B, C, at the indicated days. The secondary antibodies used were species adsorbed. Blue nuclear staining represents DAPI (Scale 10 μ m).

Question 2. Page 9. "increased concentrations of reducing sugars shifted the equilibrium towards a decreased NAD⁺ and increasing NADH cofactor pool". That high reduced sugar concentrations decrease the NAD⁺/NADH ratio is not a novel observation (*J Diabetes Res* 2014:137919, 2014). In hyperglycemic conditions, the amount of NADH produced exceeds the electron transport chain capacity leading to an increase in the NADH/NAD⁺ ratio that is known as reductive stress. Indeed, overnutrition increases H₂O₂ generation by mitochondria to relieve the reducing pressure created by fuel overload (*Trends Endocrinol Metab* 23: 142-153, 2012); an excess of NADH imposes a stress on the electron transport chain resulting in electron leakage that causes an improper O₂ reduction to superoxide anion. The cell responds by activating the transcription of antioxidants via Nrf2 activity, which increases cellular levels of GSH further contributing to reductive stress. Moreover, in hyperglycemic conditions excess glucose fluxes into the polyol pathway in which glucose is reduced by the NADPH-dependent enzyme, aldose reductase, to form sorbitol that is then converted to fructose by NAD⁺-dependent sorbitol dehydrogenase (*Diabetes* 54: 1615-1625, 2005). In a highly reduced environment (i.e., decreased NAD⁺/NADH ratio) mitochondrial ROS production is so high and fast that it overcomes the cell's ROS scavenging capacity, even though the latter should be maximally potentiated (*Biochim Biophys Acta* 1797: 865-877, 2010; *Biochim Biophys Acta* 1837: 287-295, 2014), which ultimately leads to ROS-induced cell damage (*Biochim Biophys Acta* 1847: 514-525, 2015). This suggests that reductive stress can be

detrimental by paradoxically leading to oxidative stress. The authors should take these findings into consideration when describing/discussing the results reported herein.

Response: We thank the reviewer for this great suggestion, as this suggestion not only improved the manuscript text significantly, but will also help the readers from other scientific areas. Therefore we edited the sub-section “Exposure to increasing concentration of reducing carbohydrates impair cellular DNA repair” and in “Discussion” of this manuscript and also cited the respective articles.

Question 3. Blots in Fig. EV6A (lung adenocarcinoma [A549] cells) are the same as in Fig. 2F (primary murine lung fibroblasts). A mistake during the preparation of the two Figures?

Response: We thank the reviewer for pointing out this gross, but unintentional mistake which happened during the final figure arrangements after the final suggestions from co-authors. We honestly regret this and apologize for this mistake. Below we are presenting the original figure which was got replaced with the lung fibroblast IP data figure in the old manuscript (**Figure 2R Q3**). We corrected this mistake in the revised version of this manuscript.

Figure 2R Q3: Lung adenocarcinoma (A549) cells were cultured in either low (5.5mM; lane 1) or high glucose (30mM; lane 2) for 5 days, fructose (30mM; lane 3), ribose (20mM; lane 4), for 3 days. The cell extract was then immunoprecipitated using anti-PARP, or a non-specific species control antibody. The PARP or its interacting partner DBC1 was then detected using PARP, or a DBC1 specific antibody.

Question 4. Fig. EV11C,D. Contrary to control lung, nuclei of diabetic lung are intensely positive for the DNA DSB marker pATM as are nuclei of lung of mice injected with non-phosphorylatable RAGE mutant, pointing to defective DNA repair; however injection with phosphomimetic RAGE results in a robust reduction of pATM staining. These results would put pATM downstream of phosphomimetic RAGE. However, in their previous work (Kumar, Fleming et al. 2017) the authors show that "RAGE phosphorylation by ATM is central for repair of radiation induced DNA damage and of tissue fibrosis and cancer formation" and "ATM-phosphorylated nuclear RAGE prevents cellular senescence, inflammation, pulmonary fibrosis and cancer". Thus, while ATM is required for RAGE phosphorylation and nuclear localization, transduction with phosphomimetic RAGE robustly reduces ATM activity. Can one envisage a scenario in which ATM phosphorylates RAGE and phosphorylated RAGE promote DNA repair, which leads to ATM inactivation?

Response: We thank the reviewer for this comment and for providing us the opportunity to clarify this. The accurate repair of the DNA-DSBs leads to “turn-off” of the DNA damage signaling. In particular, as shown in the figure 2R Q4, ATM serves as DNA-DSB sensor kinase and upon detecting the DSB, it gets activated (by auto-phosphorylation at Serine 1981; now known as Activated ATM or pATM). This pATM, then phosphorylates several downstream targets such as Histone H2AX, MRN complex, Checkpoint kinase 2, PARP1, RAGE and several other factors (Cimprich and Cortez 2008, Awasthi, Foiani et al. 2015). Concomitantly, these factors are recruited to the site of damage and process the broken ends of the DNA (called

end-resection) to generate the ssDNA tails. As the nucleoplasm is rich in nucleases, to prevent the non-specific degradation of these ssDNA tails, the repair machinery then coats them with ssDNA binding proteins such as RPA2 (Replication Protein-A2). RPA2, once loaded onto the DNA, it gets phosphorylated at Serine 4 & Serine 8 and this nucleoprotein complex (RPA2 onto ssDNA) now activates another DNA repair kinase ATR (gets phosphorylated; pATR) (Shiotani and Zou 2009, Kumar, Fleming et al. 2017). Activated pATR, then phosphorylates the checkpoint kinase CHK1 at S345 residue. At this stage (pATR), the cell recognizes that the DNA-DSBs repair is on track, hence Protein Phosphatases (PP family members; PP2A, PP1, Wip1, PP5 etc) dephosphorylates ATM signaling and CHK2 T68 signaling (Awasthi, Foiani et al. 2015). In short, once the DNA end-resection gets completed, cell rests ATM and CHK2 and passes the responsibility to pATR and pCHK1 for continuing the DNA repair. Furthermore, RAGE stabilizes the MRN complex onto DNA and promotes end-resection. Hence RAGE ensures the DNA-DSBs repair and thereby indirectly “Turns-off” persistent DNA damage signaling. We elaborated this signaling in the figure below (please begin from the corner; marked as Start here in red). Furthermore, in order to provide a better explanation, we edited the text and cited the respective references for readers.

Figure legend 2R Q4: Pictorial representation of the DNA-DSBs signaling and repair cascade.

Question 5. Discussion, page 15 "it remains unknown, how ROS are generated, since an increase in ROS might be caused by changes in carbohydrate metabolism (Lorenzi, Montisano et al. 1986) ". Please see point 1 above.

Response: We thank the reviewer for this suggestion, as this, as well as Q2 suggestion, not only improved the manuscript text significantly, but now will also help the readers from other scientific areas. Therefore we edited the discussion part for the ROS generation and cited the respective references in the revised version of the manuscript.

Question 6 (i). Reduction of tissue fibrosis upon transduction with phosphomimetic RAGE suggests that phosphomimetic RAGE strongly interferes with the molecular machinery that causes fibrosis. Is this a result of improved DNA repair in lung- and kidney-specific cell types exclusively or is transduced phosphomimetic RAGE also acting in macrophages (i.e. the cell type principally responsible for the liberation of fibrosis-inducing factors) and/or fibroblasts (i.e. the cell type principally responsive to fibrosis-inducing factors)? In other words, is phosphomimetic RAGE activity in lung- and kidney-specific cell types enough to improve tissue histology and function (albeit via a mechanism to be identified)?

Response: We thank the reviewer for this suggestion to analyze and clarify this. As in our AAV transduction approach, we used a pAM-CBA plasmid that has strong and constitutive promoter to express the transgene in various cells or tissues. Here we used two serotypes, AAV2 and AAV8. These serotypes can target epithelial, fibroblasts, podocytes, tubular cells, macrophages and other several other cell types too (Li, Jayandharan et al. 2010, Chung, Fogelgren et al. 2011, Payne, Takahashi et al.

2016). In short, they are highly recommended for expressing the transgene globally in lungs and kidneys (Franich, Fitzsimons et al. 2008, Li, Jayandharan et al. 2010, Payne, Takahashi et al. 2016). Further, under the hyperglycemic condition, the ROS generated affects the integrity of the genome of all the cell types representing that organ. Considering this, macrophages will also incur the DNA damage and the RAGE-EE will also promote the DNA repair in any cell types it is expressed. The RAGE-EE transduced diabetic lung and kidney show an improvement of fibrosis. The improved senescence and SASP scores are the direct and cumulative effect of improved DNA repair in all cell types of these organs infected by virions. Moreover, the activated macrophages and other immune cells have a limited lifespan; hence will clear up much earlier than the tissue's other cell types (single AAV2/8 transduction can express the desired transgene for about 40-60 weeks). But in addition, a paracrine effect of transduced cells cannot be excluded.

Furthermore, to validate our points, we used anti-CD11b, as a resident macrophage-specific marker and performed a co-staining with DNA-DSBs marker (γ H2AX) in both transduced lung and kidney. Our results show that the lung or kidney cell zone positive for CD11b (marked by yellow fluorescence) has less positive γ H2AX nuclei in the group of mice transduced with virions expressing RAGE-EE (phospho-mimetic RAGE), whereas the group which received RAGE-AA (non-phosphorylatable RAGE) or RFP alone virions, have a significant positivity for the DNA-DSBs marker along with the other cell types (Figure 2R Q6(i) A and B). The non-diabetic age-matched group served as control of the study. Considering this, we state that the observed reduction of fibrosis and SASPs, are directly associated with improved DNA-DSBs repair. Since we already included γ H2AX data in the original manuscript, we request you to consider this data shown marked as “only for the reviewer” to avoid the presentation of similar data over and over again. But considering the importance of this point, we cited the respective research articles as well as made a

clear statement that we used a global cell type target serotype of AAV virus for our transduction experiment in the result section.

Figure 2R Q6(i): **(A)** Representative images of lungs from 6-month STZ diabetic mice transduced with the indicated AAV2/8 virions, as described in Methods stained for macrophage marker CD11b and DNA-DSBs repair marker γ H2AX. The lungs were harvested 6-weeks after viral transduction and anti-RFP was used for visualizing the expression of virions. CD11b expression is indicated in Yellow. Blue nuclear staining represents DAPI. **(B)** Representative images of kidney from 6-month STZ diabetic mice transduced with the indicated AAV2/8 virions, as described in Methods stained for macrophage marker CD11b and DNA-DSBs repair marker γ H2AX. The lungs were harvested 6-weeks after viral transduction and anti-RFP was used for visualizing the expression of virions. The CD11b expression is indicated in Yellow. Blue nuclear staining represents DAPI.

Question 6 (ii). While the authors show that transduction with phosphomimetic RAGE reduces mean fluorescence intensity of the proinflammatory IL-6 in lung tissue (Fig. 5D), measurements of serum levels of proinflammatory and antiinflammatory cytokines and/or tissue levels of proinflammatory and antiinflammatory cytokine mRNAs would be advisable to support the authors' conclusion that phosphomimetic RAGE reduces inflammation.

Response: We thank the reviewer for this suggestion to analyze pro-inflammatory as well as anti-inflammatory cytokines. To answer this we used both lungs as well as kidneys from the non-diabetic control and transduced groups. The quantitative mRNA analysis of IL-6, IL-8, MCP1, TGF- β 1 and IL-1 β was performed using the primers described in Table 5.

Here it was observed that transduction of phosphomimetic RAGE mutant (RAGE-EE), but not non-phosphorylatable RAGE (RAGE-AA), can significantly decrease the mRNA expression of these cytokines (Figure 2R Q6(ii) A and B). Thus it clearly shows that the improved physiological function decreased fibrotic and senescent areas in RAGE-EE transduced organs are directly complemented by decreased inflammatory phenotype in both these organs. In the revised version of this manuscript, we included this data in figure EV15 D (lung) and EV16 D (kidney), we also edited the text, method and table section for the experimental details.

Figure 2R Q6(ii): (A) Quantitative analysis of transduction effects of RAGE (AA or EE) or RFP pro-inflammatory, fibrotic and SASP gene expression in lung tissue. The mRNA of pro-inflammatory, fibrotic and SASP cytokines was significantly suppressed in RAGE-EE transduction group. The data were normalized against control value (shown as 1) (mean \pm SD, *: $p < 0.05$, **: $p < 0.01$). (B) Quantitative analysis of transduction effects of RAGE (AA or EE) or RFP pro-inflammatory, fibrotic and SASP gene expression in kidney tissue. The mRNA of pro-inflammatory, fibrotic and SASP cytokines was significantly suppressed in RAGE-EE transduction group. The data were normalized against control value (shown as 1) (mean \pm SD, *: $p < 0.05$, **: $p < 0.01$).

References for Reviewer-2

- Awasthi, P., M. Foiani and A. Kumar (2015). "ATM and ATR signaling at a glance." *J Cell Sci* **128**(23): 4255-4262.
- Chung, D. C., B. Fogelgren, K. M. Park, J. Heidenberg, X. Zuo, L. Huang, J. Bennett and J. H. Lipschutz (2011). "Adeno-Associated Virus-Mediated Gene Transfer to Renal Tubule Cells via a Retrograde Ureteral Approach." *Nephron Extra* **1**(1): 217-223.
- Cimprich, K. A. and D. Cortez (2008). "ATR: an essential regulator of genome integrity." *Nat Rev Mol Cell Biol* **9**(8): 616-627.
- Franich, N. R., H. L. Fitzsimons, D. M. Fong, M. Klugmann, M. J. During and D. Young (2008). "AAV vector-mediated RNAi of mutant huntingtin expression is neuroprotective in a novel genetic rat model of Huntington's disease." *Mol Ther* **16**(5): 947-956.
- Kumar, V., T. Fleming, S. Terjung, C. Gorzelanny, C. Gebhardt, R. Agrawal, M. A. Mall, J. Ranzinger, M. Zeier, T. Madhusudhan, S. Ranjan, B. Isermann, A. Liesz, D. Deshpande, H. U. Haring, S. K. Biswas, P. R. Reynolds, H. P. Hammes, R. Peperkok, P. Angel, S. Herzig and P. P. Nawroth (2017). "Homeostatic nuclear RAGE-ATM interaction is essential for efficient DNA repair." *Nucleic Acids Res* **45**(18): 10595-10613.
- Li, M., G. R. Jayandharan, B. Li, C. Ling, W. Ma, A. Srivastava and L. Zhong (2010). "High-efficiency transduction of fibroblasts and mesenchymal stem cells by tyrosine-mutant AAV2 vectors for their potential use in cellular therapy." *Hum Gene Ther* **21**(11): 1527-1543.
- Payne, J. G., A. Takahashi, M. I. Higgins, E. L. Porter, B. Suki, A. Balazs and A. A. Wilson (2016). "Multilineage transduction of resident lung cells in vivo by AAV2/8 for alpha1-antitrypsin gene therapy." *Mol Ther Methods Clin Dev* **3**: 16042.
- Shiotani, B. and L. Zou (2009). "Single-stranded DNA orchestrates an ATM-to-ATR switch at DNA breaks." *Mol Cell* **33**(5): 547-558.

Thank you for submitting your revised manuscript for consideration by The EMBO Journal. Please accept my apologies for the delay in processing of your revised manuscript due to protracted referee input. Your amended study was sent back to the two referees for re-evaluation, and we have received comments from both of them, which I enclose below. As you will see the referees find that their concerns have been sufficiently addressed and they are now in favour of publication.

Thus, we are pleased to inform you that your manuscript has been accepted in principle for publication in The EMBO Journal, pending the remaining minor issues related to text editing indicated by reviewer #1, and to formatting and data representation as listed below, which need to be adjusted at re-submission.

Referee #1:

In this revised manuscript, Varun and colleagues have answered all my "questions." I find the revised manuscript much more solid, and scientifically sound. However, the text will greatly benefit from some editing to make it more fluid and easier to read. Shorter and simpler sentences could convey the same messages. This is especially true in the results section, which has numerous very long and redundant sentences. In addition, there are still numerous grammatical/English/punctuation mistakes in the text (including figure title and figure legends) that can easily be fixed with some proofreading.

Here are some examples, but there are many other

1- In the introduction, replace "This might especially be true for diabetes type-1 and type-2, which is associated with increased ROS and dicarbonyl production and thus consequently affects the integrity of the genome" by "This might especially be true for type-1 and type-2 diabetes, which are associated with increased ROS and dicarbonyl production, and thus consequently affects the integrity of the genome".

2- In the Introduction, 3rd paragraph, add "s" to organs and remove "s" from Shows (Although, organs such as kidneys, lungs and liver shows the highest incidence of fibrosis and thereby other complications".

3- Figure 1 title: remove "s" from leads "Increased pO₂, Mitochondrial Respiration and Oxidative Stress leads to the Activation of Immediate DNA Damage Response after Parturition of Normal Live Pups"

4- "A similar observation was also made in renal HEK-293 cells too (Fig EV6C, D)." Remove the word too (redundant with similar)

5- Replace in the text and in Figure EV8 "etoposid" by "etoposide"

6- As is usual, please always refer to Senescence-associated Beta-Galactosidase activity, as SA- β -Galactosidase. This is considered the norm (see PMID:20010931)

7- In the abstract (last sentence), replace "re-establish" by restore

8- In "Furthermore, NADH induction is known to shift the cellular NAD⁺/NADH equilibrium towards NADH (Ma, Chen et al. 2011, Kembro, Cortassa et al. 2014).", replace Induction by pretreatment.

9- Avoid extremely heavy and redundant sentences like "Furthermore in addition to A549 cells, these results were also confirmed in renal cells too (Fig EV8C, D)."

10- Do not duplicate sentences between the results and discussion "Thus under this highly reduced environment, mitochondrial ROS production surpasses the scavenging capacity of the cell (Aon, Cortassa et al. 2010, Kembro, Aon et al. 2013, Cortassa, O'Rourke et al. 2014), »

Referee #2:

The authors have satisfied my previous concerns.

Please see next page.

Referee #1:

In this revised manuscript, Varun and colleagues have answered all my "questions." I find the revised manuscript much more solid, and scientifically sound.

Response: We thank the reviewer for the critical and constructive evaluation of our manuscript.

However, the text will greatly benefit from some editing to make it more fluid and easier to read. Shorter and simpler sentences could convey the same messages. This is especially true in the results section, which has numerous very long and redundant sentences. In addition, there are still numerous grammatical/English/punctuation mistakes in the text (including figure title and figure legends) that can easily be fixed with some proofreading.

Here are some examples, but there are many other

1- In the introduction, replace "This might especially be true for diabetes type-1 and type-2, which is associated with increased ROS and dicarbonyl production and thus consequently affects the integrity of the genome" by "This might especially be true for type-1 and type-2 diabetes, which are associated with increased ROS and dicarbonyl production, and thus consequently affects the integrity of the genome".

Response: We thank the reviewer for this suggestion and apologize for the proofreading errors. Here in the revised version of the manuscript, we made the respective change in the introduction text of the updated version of this manuscript.

2- In the Introduction, 3rd paragraph, add "s" to organs and remove "s" from Shows (Although, organs such as kidneys, lungs and liver shows the highest incidence of fibrosis and thereby other complications".

Response: We thank the reviewer for pointing out this grammatical error and also apologize for this. We corrected this mistake in the revised version of the manuscript.

3- Figure 1 title: remove "s" from leads "Increased pO₂, Mitochondrial

Respiration and Oxidative Stress leads to the Activation of Immediate DNA Damage Response after Parturition of Normal Live Pups"

Response: We thank the reviewer for pointing out this mistake and also apologize for this. We corrected this error in the revised version of the manuscript.

4- "A similar observation was also made in renal HEK-293 cells too (Fig EV6C, D)." Remove the word too (redundant with similar)

Response: We thank the reviewer for pointing out this gross grammatical error and also apologize for this. We corrected this mistake in the revised version of the manuscript. The mentioned phrase now reads, “**Consistent results were obtained in HEK-293 cells as well (Appendix Fig S3C, D)**”. Further, due to formatting changes (from EMBO team), the figure number of EV6C, D) has been changed to Appendix Fig S3C, D.

5- Replace in the text and in Figure EV8 "etoposid" by "etoposide"

Response: We thank the reviewer for pointing out this mistake and apologize for it. We corrected it in the revised manuscript.

6- As is usual, please always refer to Senescence-associated Beta-Galactosidase activity, as SA- β -Galactosidase. This is considered the norm (see PMID:20010931)

Response: We thank the reviewer for this suggestion and made the respective change in the revised version of this manuscript.

7- In the abstract (last sentence), replace "re-establish" by restore

Response: We thank the reviewer for this suggestion. We now made the specified change in the abstract by replacing the “re-establish” with “restore”.

8- In "Furthermore, NADH induction is known to shift the cellular NAD⁺/NADH equilibrium towards NADH (Ma, Chen et al. 2011, Kembro, Cortassa et al. 2014).", replace Induction by pretreatment.

We thank the reviewer for this suggestion. We now made the suggested change in the revised version of this manuscript.

9- Avoid extremely heavy and redundant sentences like "Furthermore in addition to A549 cells, these results were also confirmed in renal cells too (Fig EV8C, D)."

Response: We thank the reviewer for pointing out this gross grammatical error and also apologize for this. The updated text now reads to "In addition, similar findings were obtained from renal cells (Appendix Fig S5C, D)".

10- Do not duplicate sentences between the results and discussion "Thus under this highly reduced environment, mitochondrial ROS production surpasses the scavenging capacity of the cell (Aon, Cortassa et al. 2010, Kembro, Aon et al. 2013, Cortassa, O'Rourke et al. 2014), »

Response: We thank the reviewer for pointing out this gross, but unintentional mistake. We now corrected this repetition error in the revised manuscript.

Referee #2:

The authors have satisfied my previous concerns.

Response: We thank the reviewer for the critical and constructive evaluation of our manuscript.

3rd Editorial Decision

8th Mar 2020

Thank you for submitting the revised version of your manuscript. I have now evaluated your amended manuscript and concluded that the remaining minor concerns have been sufficiently addressed.

Thus, I am pleased to inform you that your manuscript has been accepted for publication in the EMBO Journal.

Corresponding Author Name: Prof. Peter Paul Nawroth and Dr. Varun Kumar

Journal Submitted to: EMBO J

Manuscript Number: EMBOJ-2019-103477